Resource

# K48- and K63-linked ubiquitin chain interactome reveals branch- and length-specific ubiquitin interactors

Anita Waltho[1,3] , Oliver Popp[1], Christopher Lenz[2] , Lukas Pluska[1,3], Mahil Lambert[2], Volker Dötsch[2] , Philipp Mertins[1] , Thomas Sommer[1,3]

The ubiquitin (Ub) code denotes the complex Ub architectures, including Ub chains of different lengths, linkage types, and linkage combinations, which enable ubiquitination to control a wide range of protein fates. Although many linkage-specific interactors have been described, how interactors are able to decode more complex architectures is not fully understood. We conducted a Ub interactor screen, in humans and yeast, using Ub chains of varying lengths, as well as homotypic and heterotypic branched chains of the two most abundant linkage types—lysine 48–linked (K48) and lysine 63–linked (K63) Ub. We identified some of the first K48/K63-linked branch-specific Ub interactors, including histone ADP-ribosyltransferase PARP10/ARTD10, E3 ligase UBR4, and huntingtin-interacting protein HIP1. Furthermore, we revealed the importance of chain length by identifying interactors with a preference for Ub3 over Ub2 chains, including Ub-directed endoprotease DDI2, autophagy receptor CCDC50, and p97 adaptor FAF1. Crucially, we compared datasets collected using two common deubiquitinase inhibitors—chloroacetamide and N-ethylmaleimide. This revealed inhibitor-dependent interactors, highlighting the importance of inhibitor consideration during pulldown studies. This dataset is a key resource for understanding how the Ub code is read.

## Introduction

Ubiquitination is a post-translational modification, which regulates almost every cellular process. To achieve this, a ubiquitin (Ub) signal is added onto the substrate protein, recruiting specific ubiquitin-binding proteins (UbBPs) via their ubiquitin-binding domains (UBDs) to carry out a desired function. There are a wide range of UbBPs and functions, for example, recruitment of DNA repair proteins to the site of DNA damage (1), endocytosis adaptors binding to a membrane receptor to initiate its vesicular transport (2), or recruitment of the proteasome leading to substrate degradation (3). The building block of every Ub signal is just a simple 9.6 KD protein—Ub. How this small protein can control such a wide array of protein fates is down to the complex chain architectures that Ubs can form, known as the Ub code (4).

Substrate ubiquitination is initiated by monoubiquitination, the covalent attachment of Ub via its C-terminal carboxylate to, most conventionally, a lysine (K) of the substrate protein. This can be followed by ubiquitination of Ub itself at one of its 7 K residues or the N-terminal amide group, thus forming a Ub chain. The resulting Ub2 chain can also be described by its linkage type, the residue through which the Ub moieties are linked, for example, K48-linked Ub2 (K48 Ub2). This chain can be extended to Ub3, Ub4, and so on. Ub chains can be homotypic, meaning all Ubs in the chain are linked through the same residue, or heterotypic, in which Ubs are linked through different residues. Heterotypic chains may be mixed linkage, with alternating linkage types, or branched, where a single Ub in the chain has more than one Ub attached to it (5, 6). The Ub code encompasses this diverse range of Ub architectures, based on linkage type, chain length, and homotypic or heterotypic linkage.

K48 Ub is the most abundant linkage type in the cell, followed by K63-linked Ub (K63 Ub) (7). The former is a well-studied proteasomal degradation signal (4), and the latter is associated with pathways such as autophagy (8), protein trafficking (9, 10), and NF-κB signalling (11). Branched Ub chains containing both linkages, referred to as K48/K63-linked branched Ub (K48/K63 branched Ub), are also present in the cell, making up 20% of all K63 linkages (12). The function of this chain type is less well defined. K48/K63 branched Ub was reported in one instance to enhance NF-κB signalling (12), and in another to trigger proteasomal degradation (13). This suggests that the Ub signal may be influenced by other factors, for example, distinct architectural features of different K48/K63 branched Ub, which are currently unresolved, the cellular context of the signal, or the substrate protein. Furthermore, K48/K63 branch-specific binders are an only recently emerging area of investigation (14 Preprint, 15).

Not only linkage type, but also the length of a Ub chain can determine interactor binding. Previous Ub interactor MS screens have identified UbBPs, which only interact with long chains of methionine (M)1–linked (16), K27-linked, K29-linked, or K33-linked

[1]Max Delbrück Center for Molecular Medicine in the Helmholtz Association, Berlin, Germany   [2]Institute for Biophysical Chemistry and Center for Biomolecular Magnetic Resonance, Goethe University, Frankfurt, Germany   [3]Institute for Biology, Humboldt-University zu Berlin, Berlin, Germany

Correspondence: aswaltho@hotmail.com; tsommer@mdc-berlin.de

Ub ([17]). Some UbBPs have multiple Ub-binding sites ([18], [19]), and some deubiquitinases (DUBs) have chain length preference. For example, MINDY1 prefers longer chains ([20], [21]), whereas UCHL3 prefers shorter chains ([22]). Furthermore, it is conventionally believed that the proteasomal degradation requires conjugation of K48 ≥Ub4 ([23], [24]), although this has been contested ([25], [26]). These findings suggest that Ub chain length influences Ub binding for at least some UbBPs.

Cell-wide Ub interactor pulldown studies enable us to decode, meaning reveal the function of, Ub signals through identification of chain type–specific UbBPs. Furthermore, information on UbBP specificity aids our understanding of the mechanism of Ub binding and the role of UBDs. Thus far, published datasets have used chemically synthesized Ub chains to identify potential chain linkage– ([16], [27]), chain length– ([17]), and branch-dependent ([15]) Ub interactors in humans. Our dataset builds on this information using native enzymatically synthesized Ub chains to probe for linkage-, chain length–, and branch-specific interactors of K48 and K63 Ub in both humans and budding yeast. We identified interactors with a preference for Ub3 over Ub2, including CCDC50, FAF1, DDI1, and its yeast homologue Ddi1, and K48/K63 branch-specific interactors, including PARP10, UBR4, and HIP1. We were able to validate HIP1's K48/K63 branched Ub preference by surface plasmon resonance (SPR). Furthermore, we investigated, by comparison, the effect of reagents commonly used as DUB inhibitors, N-ethylmaleimide (NEM), and chloroacetamide (CAA), on Ub binding.

# Results

### Ubiquitin interactor screen establishment

We designed a K48 and K63 Ub interactor screen in which Ub chains are immobilized on resin and used as bait to enrich Ub interactors from the cell lysate. Interactors are then identified by liquid chromatography–mass spectrometry (LC-MS) and chain-type enrichment patterns analysed by statistical comparison ([Fig 1A]). As we are interested in comparing chain linkage–, length-, and branch-specific Ub interactors, we sought to synthesize mono-Ub, homotypic K48 and K63 Ub2 and Ub3, and K48/K63 branched Ub3, subsequently referred to as Br Ub3. We chose to use Br Ub3 as it encompasses the branchpoint, the basic unit of a branched chain. As such, it forms the basis of the complex architecture of branched chains in the cell, which is not fully elucidated and may vary in different contexts. We previously discovered the K48-branching activity of the Ub-conjugating (E2) enzyme Ubc1 ([28]). Using this enzyme, along with K48- and K63-specific E2 enzymes, CDC34 and Ubc13/Uev1a, we were able to enzymatically synthesize and purify our desired Ub chains in vitro. Chain linkage composition was confirmed using the UbiCRest method ([29]) by selective disassembly with the K48- and K63-specific DUBs OTUB1 and AMSH, respectively ([Fig S1A]).

To immobilize the Ub chains on streptavidin resin, we added a serine/glycine repeat linker containing a single cysteine residue after the C-terminus of the proximal Ub of each chain and subsequently attached a biotin molecule using a cysteine–maleimide reaction. Complete biotin conjugation was confirmed using intact

MS ([Fig S1B–G]). These Ub chains contain native isopeptide bonds; therefore, they are susceptible to chain disassembly by endogenous DUBs in the lysate. As cysteine proteases are the largest DUB family, cysteine alkylators including CAA and NEM are often used as DUB inhibitors ([29], [30], [31], [32]). However, cysteine alkylators do not just target DUBs, but they also can theoretically alkylate any exposed cysteine on a protein. Furthermore, although CAA is relatively cysteine-specific ([33]), NEM, when used for peptide alkylation for MS, was found to have frequent side reactions with N-termini and K side chains ([34]). Off-target effects on non-DUB proteins are a concern as they can alter Ub-binding surfaces. For example, NEM combined with iodoacetamide (IAA) treatment was found to perturb NEMO binding to K63 Ub chains in vitro ([35]).

We tested CAA and NEM for their ability to stabilize immobilized Ub chains in the HeLa cell lysate ([Fig 1B]). We found that the anti-Ub antibody showed some linkage-dependent binding ([Fig S10A]). Known linkage-specific UbBPs, K48-specific RAD23B ([36]) and K63-specific EPN2 ([16]), were used as positive controls for selective UbBP enrichment. With either CAA or NEM, RAD23B and EPN2 were only enriched on their preferred linkage types, showing that both inhibitors block chain disassembly sufficiently for specific UbBP pulldown. However, there were differences in the stability of immobilized Ub in each inhibitor-treated lysate; in NEM, there was nearly no chain disassembly, whereas in CAA, Ub3 was partially disassembled to Ub2 ([Fig 1B]). This could be expected as NEM is a more potent cysteine alkylator ([30]). Whilst acknowledging the limitation of partial digestion under CAA treatment, it is notable that the original bait remains the predominant Ub species throughout the experiment. Moreover, the amount of Ub bait added significantly exceeds that of the individual proteins in the lysate. Taken together with the distinct binding patterns of the known UbBPs, we suggest that the CAA approach is effective in selectively enriching chain-specific UbBPs, despite the partial chain digest. Taking into account the discussed advantages and disadvantages of using either CAA or NEM, we chose to perform the Ub interactor screen with each inhibitor separately. By comparison of these two datasets, we could identify overlapping and inhibitor-specific Ub interactors, and thus assess the suitability of each inhibitor for protein interaction studies.

### Ubiquitin interactor enrichment patterns

Following our established set-up, Ub interactors were enriched from the CAA- or NEM-treated human HeLa or CAA-treated yeast Saccharomyces cerevisiae cell lysate and identified by LC-MS ([Fig 1A]). Principal component analysis showed clustering of samples by bait type ([Fig S2A and B]). After filtering, normalization, and imputation, 4,540 and 4,526 unique protein isoforms were identified across pulldowns from the CAA- and NEM-treated HeLa lysate, respectively, with an overlap of 3,889. We compiled a list of expected UbBPs by combining proteins found under the Gene Ontology term Ub-binding 0043130 and UBD-containing proteins from the integrated annotations for Ubiquitin and Ubiquitin-like Conjugation Database (iUUCD) ([38]) and literature research (Table S1, s5 and 6). 242 of the total identified protein isoforms in the CAA dataset were expected UbBPs. In the NEM dataset, this number was 214 ([Fig S3A]).

To remove unspecific background binders and interrogate chain type–dependent enrichment patterns, proteins were prefiltered by

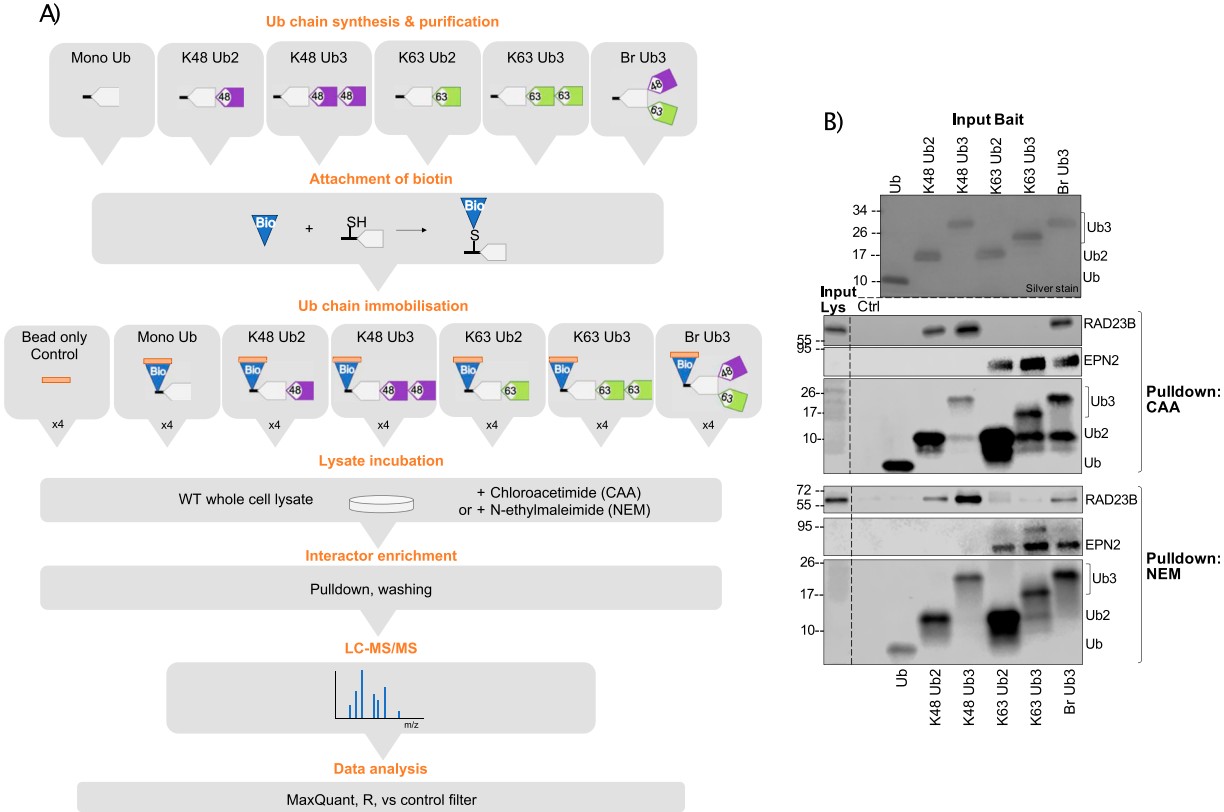

**Figure 1. Ubiquitin chain interactor enrichment using different deubiquitinase inhibitors.**
**(A)** Schematic of the ubiquitin (Ub) interactor enrichment method. In short, Ub chains are enzymatically synthesized, purified, and conjugated to biotin. Ub chains are immobilized on streptavidin resin and incubated with chloroacetamide- or N-ethylmaleimide–treated cell lysate in quadruplicate. Pulldown with bead only is used as a negative control for background binders. Interactors are enriched and unspecific binders washed off. Enriched interactors are identified by LC-MS/MS and analysed using MaxQuant and R, including a prefiltering step to select for proteins enriched on Ub over the bead-only control. **(B)** Western blot of Ub interactor pulldown using chloroacetamide or N-ethylmaleimide, as deubiquitinase inhibitors. Silver stain of input Ub. Pulldown blotted using anti-RAD23B, anti-EPN2, and anti-Ub antibodies. Input lys is input lysate. Ctrl is bead-only control pulldown.

significant enrichment in any Ub chain pulldown compared with the bead-only control (two-sample moderated *t* test, adjusted *P*-value [Adj.*P*] < 0.05, log(fold change (FC)) > 0). Prefiltering significantly increased the proportion of expected UbBPs to 122/544 and 76/206 protein isoforms in CAA and NEM datasets, respectively (Fisher's exact test, $P < 2.2 \times 10^{-16}$), suggesting that our prefiltering method positively selects for UbBPs (Fig S3B). In comparison with a published HeLa global proteome (37), expected UbBPs were enriched in the prefiltered protein list by a factor of 11.68 for CAA and 19.22 for NEM. 114 and 72 of the protein isoforms selected by prefiltering contained known UBDs in the CAA and NEM datasets, respectively (Fig S3B, Table S1, s1 and 2). These findings support the efficacy of our method for enriching UbBPs. We compiled a summary table of all subsequent statistical comparisons combined with known UBD information for all prefiltered protein isoforms from CAA- and NEM-treated human and yeast datasets (Table S1, s1–4).

To identify chain type–specific enrichment patterns, we compared interactomes across all Ub chain types generating 286 and 139 significant differently enriched proteins in CAA and NEM datasets, respectively (moderated F test, Adj.*P* < 0.05). This included 95 expected UbBPs with CAA and 64 with NEM (Fig 2A). Comparison with a published HeLa global absolute proteome (37) revealed that

our significant interactors are not biased towards highly abundant proteins (Fig S2D and E). Significant interactors for each chain type were well correlated between NEM and CAA datasets, especially for K48/K63 branched chain interactors (Fig 2B).

We next sought to identify DUB inhibitor–dependent patterns, by comparing the enrichment across chain types of significant hits from either dataset (Fig S4A–F). A number of 50 significant hits in the CAA dataset were not identified in any Ub pulldown from the NEM-treated lysate, including 10 expected UbBPs (Fig S4C and F), whereas 11 significant hits in the NEM dataset were not identified in any Ub pulldown from the CAA-treated lysate (excluding multiple isoforms) (Fig S4E). Most significant hits shared across datasets, which were also expected UbBPs, had the same enrichment pattern in both datasets (Fig S4A). However, IKBKG/NEMO, an expected K63-specific UbBP, was significantly enriched on K63 Ub in CAA and on K48 Ub in NEM. This observation was previously reported in vitro (35). Several other common significant expected UbBPs, including MINDY3, BIRC2, GGA3, DNAJB2, XIAP, and TSG101, exhibited a preference for K63 Ub with CAA and a more general preference for both K48 and K63 Ub3 with NEM. A similar pattern of increased enrichment on K48 Ub3 in the NEM condition was also observed for several common significant hits, which were not expected UbBPs: KATNAL2, STON2,

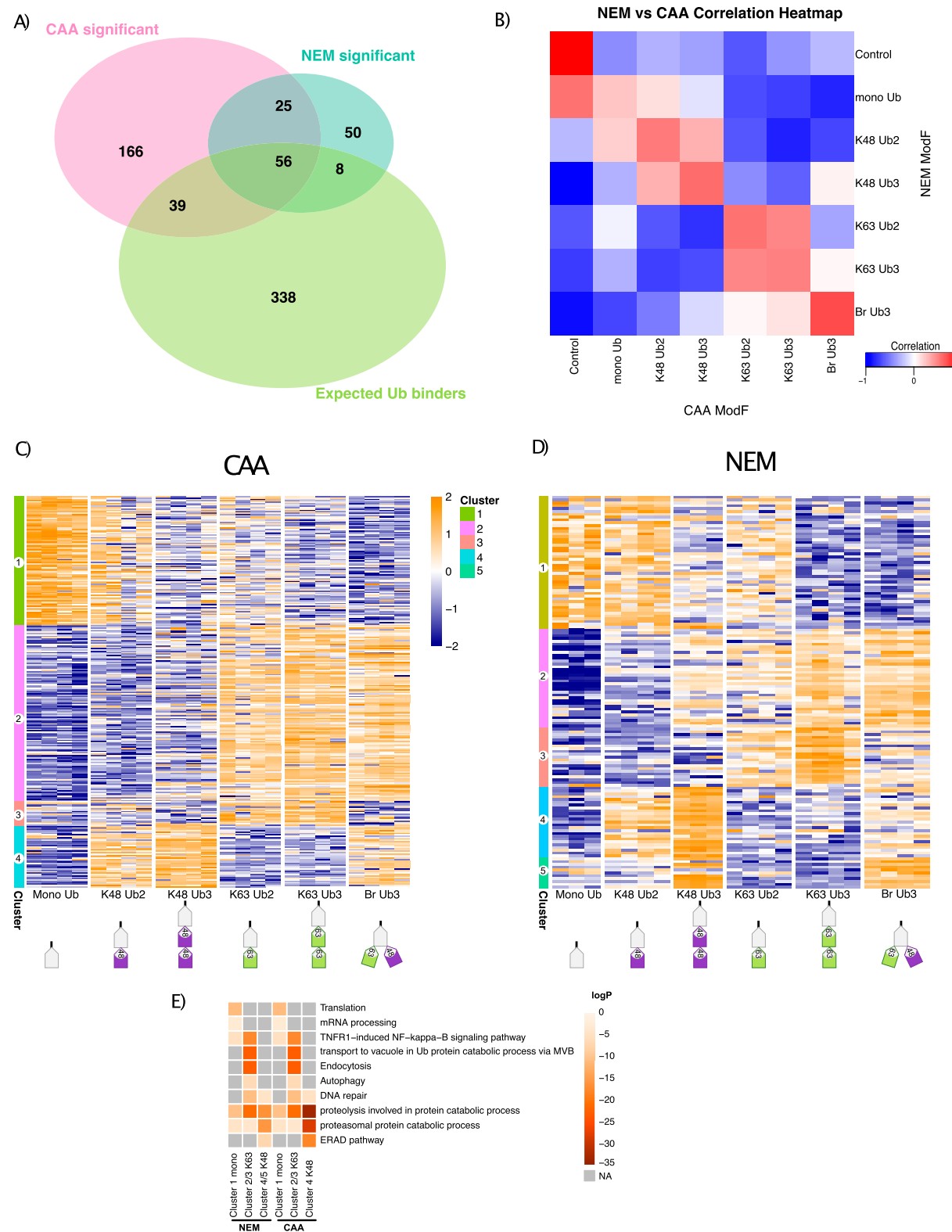

**Figure 2. Comparison of interactor enrichment patterns between chloroacetamide (CAA) and N-ethylmaleimide (NEM) datasets.**
**(A)** Overlap of significant differently enriched interactors across Ub pulldowns from the CAA- and NEM-treated lysate with expected Ub binders (expected UbBPs). Data were prefiltered by Ub enrichment (significant enrichment on Ub over bead-only control in at least one two-sample moderated *t* test comparison, log(FC) > 0, Adj.*P* < 0.05). Significant differently enriched interactors were identified from prefiltered Ub-enriched interactors by a moderated F test (Adj.*P* < 0.05). The expected Ub binder list was compiled from the Gene Ontology term Ub-binding 0043130 and UBD-containing proteins from the iUUCD ([38](http://iuucd.biocuckoo.org/)) (http://iuucd.biocuckoo.org/) and literature research.

UBFD1, NIPSNAP2, LACTB, and NIPSNAP1 (Fig S4B). These observations may be a result of increased chain stability by the more potent DUB inhibitor NEM or unspecific alkylation by NEM affecting Ub-binding sites, as is the case for IKBKG/NEMO (35).

Significant proteins clustered into four or five similar clusters for CAA and NEM, respectively: proteins significantly enriched on mono-Ub and Ub2 (Cluster 1), on K63 Ub (Clusters 2 and 3), or on K48 Ub (Clusters 4 and 5) (Fig 2C and D, more detailed in Figs S5 and S6). In Cluster 1, there was an overlap of only four proteins between datasets: known UbBP autophagy receptor SQSTM1/p62, mitochondrial matrix protein NIPSNAP2, arginine methyltransferase PRMT5, and mitochondrial inner membrane space serine protease LACTB (Fig S7A). Interestingly, NIPSNAP2 and SQSTM1/p62 have been reported to interact with each other during the initial stages of mitophagy (40). Gene Ontology (GO) enrichment analysis revealed an enrichment of translation and mRNA processing–related proteins in Cluster 1 in both datasets (Fig 2E).

There were 42 common hits in Clusters 2 and 3, proteins with a binding preference for K63 Ub, in both datasets. This included known K63-specific UbBPs: endosomal sorting complex required for transport (ESCRT) proteins STAM (41), STAM2, and TOM1 (42), endocytic adaptor proteins ANKRD13A, ANKRD13B, and ANKRD13D (43), BRCA1-A subunit UIMC1 (44, 45), IL-1 signalling–related protein TAB2 (46), and branching E3 ligase HUWE1 (12) (Fig S7D). Notably, K48-processing DUB MINDY3 was also in Cluster 2. This aligns with the recent finding that MINDY3 prefers cleaving K48 Ub within a (K48)/K63 branched Ub4 chain (14 *Preprint*). GO enrichment revealed that proteins in Clusters 2 and 3 were associated with NF-κB signalling, Ub-dependent vacuolar transport, endocytosis, and autophagy (Fig 2E), in line with current literature on K63 Ubs (8, 9, 10, 11). Most proteins with a preference for K63 linkages were equally enriched on homotypic K63 and Br Ub3 (Cluster 2) (Fig S7B). However, there was also a cluster of K63 Ub-specific proteins, which were more enriched on homotypic K63 than Br Ub3 (Cluster 3), including, conserved across datasets, ESCRT component Tom1 (Fig S7C).

Proteins enriched on K48 Ub (Cluster 4 in the CAA dataset and Clusters 4 and 5 in the NEM dataset) included known K48-specific UbBPs: proteasomal shuttle factors RAD23A (47) and RAD23B (36), VCP/p97 adaptors UFD1 (48) and FAF1 (49), and DUBs ATXN3 (45, 50), OTUD5 (51), and MINDY1 (20) (Fig S7E). As expected from the literature, proteins in these clusters were strongly associated with proteasomal degradation and endoplasmic reticulum–associated degradation (ERAD) (4) (Fig 2E). Strikingly, in the NEM dataset, interactors enriched on K48 Ub were separated into two clusters: proteins with a preference for homotypic K48 Ub3 (Cluster 4) and those either with a Br Ub3 preference or equally enriched on both homotypic K48 and Br Ub3 (Cluster 5) (Fig S6). The former (Cluster 4) includes proteasome regulatory subunit PSMD4, proteasomal shuttle factors RAD23A and RAD23B, DUB

MINDY2, and VCP/p97 adaptor UBXN1. The latter (Cluster 5) includes DUB MINDY1, which selectively cleaves long K48 Ub chains (52), but was also recently found to preferentially cleave heterotypic K48/K63 Ub (14 *Preprint*).

The clustering results described here support the quality of our data as the chain preference of known linkage-specific UbBPs and enrichment of linkage-specific pathways are reproduced. Furthermore, they provide Ub linkage–specific binding patterns for potential novel UbBPs or UbBPs whose chain preference was previously unknown.

Our yeast Ub interactor screen identified 2,315 unique protein isoforms after filtering, normalization, and imputation, including 76 expected UbBPs (Fig S3C). Prefiltering for Ub-enriched proteins (described above) resulted in 247 proteins, including 38 expected UbBPs of which 37 contain known UBDs (two-sample moderated $t$ test, Adj.$P < 0.05$) (Fig S3D). The proportion of expected UbBPs in the prefiltered protein list was significantly increased in comparison with the expected UbBPs in the total proteins identified (Fisher's exact test, $P < 2.2 \times 10^{-16}$). We compared interactor enrichment across Ub chain types generating 148 significant differently enriched proteins (moderated F test, Adj.$P < 0.05$) (Fig S3E). Significant proteins clustered into three clusters: proteins enriched significantly on mono-Ub and partially on K63 Ub2 (Cluster 1), on K63 Ub (Cluster 2), or on K48 Ub (Cluster 3) (Fig S8). Cluster 1 included expected UbBPs with known UBDs, but whose chain-type specificity was unknown, for example, Prp-containing Duf1 and UBA-containing Gts1 (53, 54) (Table S1, s3).

Amongst interactors enriched on K63 Ub (Cluster 2) were known K63-specific UbBPs including endocytic regulator Ent2 (55), clathrin adaptor Gga2 (55), and ESCRT components Vps27 and Hse1 (55). Cluster 2 also contained UBD-containing UbBPs whose chain-type specificity was previously unknown, including LSB5, which contains GAT and VHS domains, and Rsp5 cofactor Rup1, which has a UBA domain (56) (Table S1, s3). K48 Ub-enriched proteins (Cluster 3) included known K48-specific UbBPs, for example, proteasomal receptor Rpn10 (55), Cdc48 adaptor proteins Npl4 (57) and Shp1 (55), and proteasomal shuttle protein Rad23 (55). Expected UbBP Cia1, which has a Prp UBD and for which chain-type specificity was previously unknown, was also in Cluster 3 (Table S1, s3).

We validated the chain type–specific pulldown of several known UbBPs: Vps9, Dsk2, Rad23 (55), Ddi1 (58), Yuh1, and Npl4 (57) by Western blot (Fig S10C). Overall, the yeast dataset successfully reproduced the chain preference for known linkage-specific UbBPs, whilst also providing chain-type specificity information for UbBPs with previously unknown preference. Nevertheless, it is important to consider that indirect Ub interactors, such as those belonging to large Ub-binding complexes, can also be identified using our method. In yeast, this was especially apparent as proteasomal

---

**(B)** Correlation of bait interactomes between CAA and NEM datasets. The Spearman correlation was calculated by comparing the moderated F values of common significant differently enriched proteins in each pulldown between datasets. A moderated F test before prefiltering (Adj.$P < 0.05$). **(C, D)** Clustering of significant differently enriched proteins from (C) CAA and (D) NEM datasets. **(A)** Significance determination as in (A). Hierarchical clustering by the Euclidean distance. Heatmap of iBAQ values, scaled by row using z scoring. Large-scale version of heatmaps including protein row names in Figs S5 and S6. **(C, D, E)** Gene Ontology (GO) enrichment heatmap of clusters from (C, D). GO enrichment was calculated with Metascape (39) (http://metascape.org), with a minimum overlap of 2, $P$-value cut-off < 0.01, and minimum enrichment of 1.5. Grey is not enriched. Some GO terms are abbreviated: TNFR1–induced NF-kappa–B is TNFR1–induced NF–kappa–B signalling pathway, transport to the vacuole in Ub protein catabolic process via MVB is protein transport to the vacuole involved in ubiquitin-dependent protein catabolic process via the multivesicular body sorting pathway, and ERAD is endoplasmic reticulum–associated protein degradation.

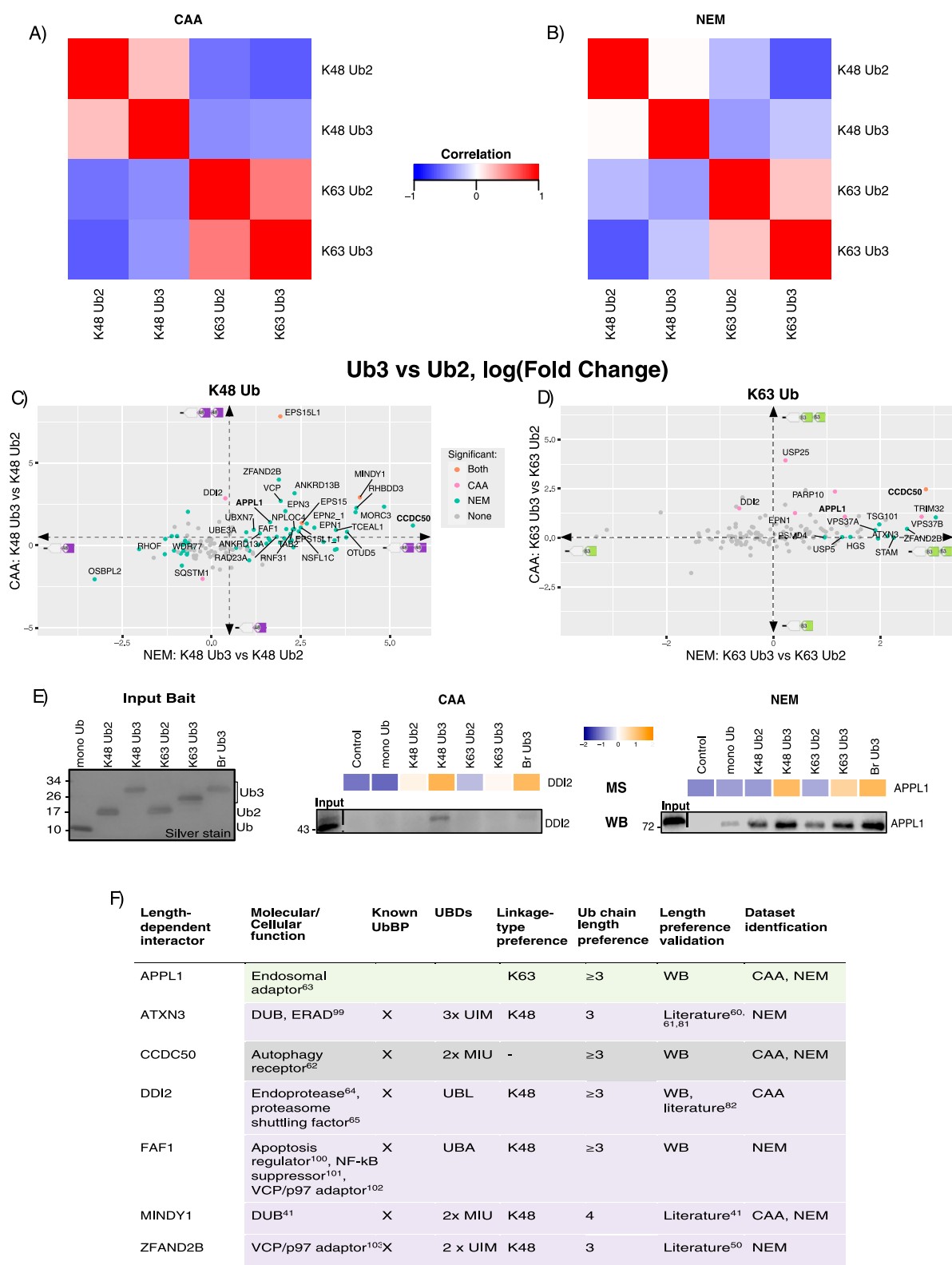

**Figure 3. Ubiquitin chain length–dependent interactors.**
**(A, B)** Correlation of homotypic Ub chain interactomes within (A) chloroacetamide (*CAA*) and (B) N-ethylmaleimide (*NEM*) datasets. The Spearman correlation was calculated using the moderated F values for each Ub pulldown of significant differently enriched interactors. Significant differently enriched interactors were identified by a moderated F test of prefiltered Ub-enriched proteins (Adj.$P$ < 0.05). **(C, D)** Scatterplot of Ub3 versus Ub2 interactor enrichment. Comparison of K48 Ub3 versus K48 Ub2 in (C) and K63 Ub3 versus K63 Ub2 in (D), by a two-sample moderated $t$ test. Comparison from CAA data plotted on the y-axis and from the NEM dataset plotted on the

subunits Rpt1, Rpt2, Rpt3, Rpt4, Rpt5, Rpt6, Rpn2, Rpn3, Rpn5, Rpn6, Rpn7, Rpn8, Rpn9, Rpn12, and Cdc48 and its adaptors Ufd1 and Ubx5 were enriched on K48 Ub (Cluster 3), despite not directly binding to Ub (55, 59, 60) (Fig S8).

### Length-dependent Ub interactors

The effect of chain length on UbBP binding is understudied. A recent ubiquitome from yeast reported that K63 linkages mainly exist as Ub2 and K48 as Ub3 or Ub4 (61). This finding suggests that endogenous Ub chains may be shorter than previously thought. It is therefore of interest to study whether Ub interactors are influenced by chain length in short chains.

With this in mind, we decided to compare the interactomes of homotypic K48 and K63 Ub2 and Ub3 chains. In all three datasets, there was a stronger correlation between the K63 Ub2 and Ub3 interactomes than between K48 Ub2 and Ub3 interactomes (Figs 3A and B and S9C). This observation was more apparent in the NEM dataset compared with the CAA dataset. Furthermore, with NEM, there were 99 and 13 significant differently enriched proteins in a Ub3 versus Ub2 comparison for K48 and K63 Ub, respectively, whereas with CAA, only 20 and 10 interactors were enriched on Ub3 in the same comparisons (moderated *t* test, Adj.*P* < 0.05). This could be the result of Ub3 disassembly to Ub2 in the CAA-treated lysate pulldown (Fig 1B). In both datasets, the mono-Ub interactome was most well correlated with the K48 Ub2 interactome (Fig S9A and B). Taken together, these results suggest that the effect of chain length on UbBP specificity varies depending on the linkage type.

In a pairwise comparison, ESP15, ESP15L1, and MINDY1 were significantly enriched on K48 Ub3 over Ub2 in both datasets. In the same comparison, 27 additional proteins were significantly enriched on Ub3 exclusively with NEM. Ub3 preference for these proteins was conserved with CAA despite not meeting the significance cut-off (two-sample moderated *t* test, log(FC) > 0.5, Adj.*P* < 0.05) (Fig 3C and F). Multiple known K48-specific UbBPs were amongst these 27, including RAD23A (47), FAF1 (49), ZFAND2B (18), and DUBs ATXN3 (45), MINDY1 (20), and OTUD5 (51). ZFAND2B, ATXN3, and MINDY1 were previously shown to bind longer chains (18, 20, 62, 63), thus supporting our findings. Unexpectedly, some interactors enriched on K48 Ub3 over Ub2 are known K63-specific UbBPs, including ANKRD13A, ANKRD13B (43), EPN1, EPN2 (16), and TAB2 (46). This observation is in line with a previous finding that a K63-specific UBD also bound longer K48 Ub chains by avid binding to non-adjacent Ub moieties (45). Moreover, it was shown in yeast that some K63-specific UbBPs can bind longer K48 Ubs (55). SQSTM1/p62, OSBPL2, RHOF1, and WDR77 were significantly enriched on K48 Ub2 over Ub3 in the NEM dataset, with the same Ub2 preference in CAA, but above the significance cut-off (two-sample moderated *t* test, log(FC) < 0.5, Adj.*P* < 0.05) (Fig 3C).

In the K63 Ub pairwise length comparison, only autophagy receptor CCDC50 (64) was significantly enriched on Ub3 over Ub2 in both datasets. ESCRT-I components TSG101 and VPS37B were significantly enriched on Ub3 in the NEM dataset only. Endosomal adaptor APPL1 (65), E3 ligase TRIM32, and ADP-ribosyltransferase PARP10 were significantly enriched on Ub3 in CAA only. All of these proteins, however, had conserved Ub3 preference between datasets, despite not meeting the significance cut-off (two-sample moderated *t* test, log(FC) > 0.5, Adj.*P* < 0.05) (Fig 3D and F).

We also compared chain length–dependent interactor enrichment within each DUB inhibitor–treated dataset (Fig S9D and E). In the CAA dataset, only Ub-directed endoprotease DDI2 (66) had significant preference for both K48- and K63 Ub3 compared with their Ub2 counterparts (Figs 3C and D and S9D). By Ub interactor pulldown and Western blot, we were able to validate DDI2's preference for K48 Ub3 over Ub2 in the CAA-treated lysate and APPL1's preference for K48 and K63 Ub3 over Ub2 in the NEM-treated lysate (Fig 3E). In yeast, the only interactors enriched on Ub3 over Ub2, independent of the linkage type, were the yeast homologue of DDI2 Ddi1 and Pmt4 (Fig S9F). We validated the Ub3 preference for Ddi1 by the Ub pulldown and Western blot (Fig S10C).

Finally, we sought to expand our length preference study with Ub4 (Fig S1H and I). Difficulty equalizing the immobilized Ub inputs and Ub4 disassembly in the lysate led to less Ub4 than other chain types in the Ub interactor pulldown (Fig S11A and B). Thus, it was difficult to elucidate Ub4 binding preference as interactors may have appeared less enriched on Ub4 for these reasons. Nonetheless, using the Ub interactor pulldown and Western blot, we were able to observe K48 Ub3 over Ub2 preference for FAF1 and DDI2, and K63 Ub3 over Ub2 preference for CCDC50 with either inhibitor. Again, for APPL1, Ub3 over Ub2 preference, independent of the linkage type, was only observed with the NEM lysate (Fig S11B). In summary, a comparison of Ub2 versus Ub3 interactomes revealed that the effect of chain length on interactor binding is linkage-dependent, with more interactors significantly enriched on Ub3 over Ub2 in K48 than K63 Ub comparisons.

### K48/K63 branched chain interactors

In this screen, we also investigated cell-wide K48/K63 branched Ub chain interactors. Br Ub3 contains three Ub moieties and three isopeptide bonds like a homotypic Ub3; however, it only contains one isopeptide bond of each linkage type like a homotypic Ub2. For this reason, we chose to compare Br Ub3 with both homotypic K48 and K63 Ub2 and Ub3.

In yeast, the Br Ub3 interactome correlated most with the K48 Ub3 interactome, whereas in the HeLa CAA dataset, it correlated most with K63 Ub3. In the HeLa NEM dataset, the correlation between branched interactome and K48 or K63 Ub3 interactomes was relatively comparable (Fig S9A–C).

---

x-axis. Dot colours indicate significance: orange is statistically significant in both datasets, pink is significant in the CAA dataset only, teal blue is significant in the NEM dataset only, and grey is not significant in either dataset. Labelled proteins were significant in at least one dataset and have consistent fold change directions in both datasets: log(FC) < −0.5 or > 0.5 for K48, and log(FC) < 0 or > 0 for K63 (as indicated by the dotted line arrows). DDI2 is labelled despite log(FC) = 0.39 with NEM because of interest based on the same length preference for yeast homologue Ddi1. Adj.*P* < 0.05. **(E)** Heatmap based on MS data and Western blot validation of length-dependent enrichment of selected interactors. DDI2 enriched from the CAA-treated lysate (left) and APPL1 enriched from the NEM-treated lysate (right). Heatmap using moderated F values (moderated F test of prefiltered Ub-enriched proteins, Adj.*P* < 0.05). The same Ub interactor pulldown experiment as in Fig 1B, and thus the same silver stain of input Ub. Pulldown blotted using anti-APPL1 and anti-DD2 antibodies. **(F)** Table of identified Ub chain length–dependent interactors. Linkage-type preference was identified from a two-sample moderated *t* test of K48 Ub3 versus K63 Ub3. Adj.*P* < 0.05. **(F)** Chain length preference validated by Western blot, (F) or (Fig S11), or literature search.

We observed that the Br Ub3 shares many interactors with K48 and K63 Ub, including multiple known linkage-specific UbBPs (Figs 2C and D and 4A–D). This suggests that a branchpoint can act as a combination of both of its constituent Ub signals.

Ub binding appeared to be influenced by both homotypic versus branched preference and chain length preference. There was an overlap of proteins significantly enriched on Br Ub3 over homotypic Ub2 and proteins enriched on homotypic Ub3 over homotypic Ub2, including DDI2, APPL1, FAF1, CCDC50, ATXN3, and MINDY1 (two-sample moderated *t* test, Adj.*P* < 0.05) (Fig 4A and B). In the NEM dataset, some known K48-specific UbBPs, including FAF1 (49), UFD1 (48), and ZFAND2B (18), and K63-specific UbBPs, including STAM (41) and HUWE1 (12), had a preference for Br Ub3 over Ub2 of their preferred linkage. However, in a pairwise comparison with homotypic Ub3, known linkage-specific UbBPs were rather enriched on homotypic Ub3 over Br Ub3 (Fig 4C and D).

ADP-ribosyltransferase PARP10 and E3 ligase UBR4 were significantly enriched on Br Ub3 compared with both homotypic K48 and K63 Ub2 and Ub3 in both datasets (Fig 4A–E). PARP10 contains two UIMs and was previously found to bind K63 ≥Ub4 (67) or K48 Ub2 (16). UBR4 plays a role in the formation of K11/K48-linked (68, 69) and K48/K63 branched Ub chains (13). In the CAA dataset, endocytosis regulator HIP1 (70), CUL3 ligase adaptor ANKFY1 (71), DUB USP48, E3 ligase MIB2 (72), DNA polymerase accessory factor RFC1 (73), and protein of unknown function BMERB1 were additional significant Br Ub3-enriched interactors in comparison to both homotypic Ub3 linkage types (Fig 4C and E). With NEM, we additionally identified transcriptional regulator MORC3 (74, 75) and lysosomal trafficking factor GGA3 (76) as significant Br Ub3 enriched in the same two-way comparisons (Fig 4C–E). Notably, RFC1 and MORC3 were also identified as K48/K63 branched Ub chain-specific interactors in a recent preprint (14 *Preprint*).

We also identified proteins with a preference for homotypic chains compared with the branched chain (two-sample moderated *t* test, Adj.*P* < 0.05). Both RAD23A and RAD23B were consistently significantly enriched on homotypic K48 Ub over Br Ub3 in both datasets. In NEM, TAB1, ACAD11, and ITSN2, and in CAA, TOLLIP and MAP3K7 were significantly enriched on homotypic K63 Ub over Br Ub3 (Fig 4A–D). In addition, the DUB USP11 was significantly enriched on all other Ub chains over Br Ub3 in the CAA dataset (Fig 4A and C). USP11's enrichment pattern was validated by Western blot (Fig S10B).

We chose to further investigate two of the interactors enriched on Br Ub3: PARP10, which was conserved across datasets, and HIP1, which was only found with CAA treatment. Ub chain pulldown with Western blot validated our MS results (Fig 5A and B). In the NEM-treated lysate, HIP1 was not sufficiently enriched in any Ub or control pulldown in line with our MS data. For PARP10, we also included Ub4 chains in the Western blot pulldown, as literature suggests that PARP10 could bind K63-≥Ub4 (67). To further validate HIP1's Br Ub3-binding specificity, we conducted SPR with different Ub chains and the immobilized Ub-binding ANTH domain of Hip1 (22-309) (77). We calculated $K_D$ values using the steady-state affinity model (Figs 5C and D and S12A–G). We determined a $K_D$ value of 0.07 μM for Br Ub3 compared with 1.55 μM for K63 Ub3, 1.89 μM for K63 Ub2, and 196.73 μM for mono-Ub. For K48 Ub2 and Ub3, we predict $K_D$ values of 137.47 μM and 66.2 μM, respectively; however, the binding affinity was outside of the concentration range tested; therefore, these values may be less accurate. These results validate Hip1 as a K48/K63 branched Ub-specific UbBP.

## Discussion

Our dataset provides a resource of K48 and K63 Ub interactors in humans and budding yeast, and their chain length and homotypic versus branched heterotypic binding preference. The linkage-specific interactors we identified included many known K48- and K63-specific UbBPs and their associated cellular pathways. Also, amongst the interactors identified with a binding preference for Ub3 over Ub2 were UbBPs with multiple Ub-binding sites, which are known to bind ≥Ub3. Furthermore, we validated the K48/K63 branched Ub preference for one of our MS hits, HIP1, using SPR binding assays. Taken together, these results validate our method's ability to identify chain type–specific interactors.

The length of Ub chains in the cell and the effect this has on Ub binding are still being elucidated. Interestingly, our data revealed that chain length (Ub3 versus Ub2) has a larger effect on K48 Ub binding than on K63 Ub binding. This observation could be due to the topologies of different chain linkage types. K48 Ub moieties self-associate and have a higher propensity to form more "closed" structures (78, 79, 80), whereas K63 Ub chains have a more open "beads-on-a-string" topology (80, 81). Thus, the conformational space of K48 Ub may be more affected by additional Ub moieties (82). In addition, it was shown in yeast that K63 linkages exist most often in Ub2 chains (61), which could reduce the need for K63-specific UbBPs with preference for longer Ub chains, whereas K48 linkages were found to exist in both Ub2 and ≥Ub3 chains. We also observed that K63-specific UbBPs were more enriched on Ub3 K48 Ub chains than K48 Ub2. This could be a result of avid binding to non-adjacent Ub moieties (45) or the wider conformational distribution of K48 >Ub2 (82). Interactors enriched on K48 Ub3 over K48 Ub2 included UbBPs with previously reported specificity for longer Ub chains: ZFAND2B/AIRAPL (18), ATXN3 (62, 63, 83), and MINDY1 (20), which bind 3, 3, and 4 Ub moieties, respectively, and DDI2/Ddi1 in yeast (58). We also identified Ub3 preference for proteins with previously undetermined chain length specificity: CCDC50, APPL1, and FAF2. The molecular mechanisms behind their length preference and the relevance of this for their cellular function are a direction for future study.

Elucidating the interactors of branched Ub chains is a key area of investigation in the Ub field. We observed that linkage-specific Ub interactors also bind K48/K63 branched chains. This supports the model of branched chains as a combination of two Ub signals. Contrastingly, it has been suggested that branching can inhibit UbBP binding, as reported for K63-specific DUB CYLD (12). In different statistical comparisons, we observed significant homotypic chain preference for known K63-specific UbBP TOM1 and K48-specific UbBPs RAD23A and RAD23B, across datasets, and DUB USP11, in the CAA dataset alone. These observations suggest that in some contexts, a branch may reduce Ub binding.

The cellular functions of branched Ub interactors can reveal the pathways in which branched Ub chains play a role. Two of the Br Ub3-enriched interactors identified here, RFC1 and MORC3, were also found as K48/K63 branch-specific in a recent preprint (14 *Preprint*). Interestingly, RFC1, along with two other proteins we identified as Br Ub3-specific—PARP10 and USP48—, is implicated in DNA damage repair processes (84, 85, 86). Supporting this, in yeast we observed sensitivity to the DNA-damaging agent

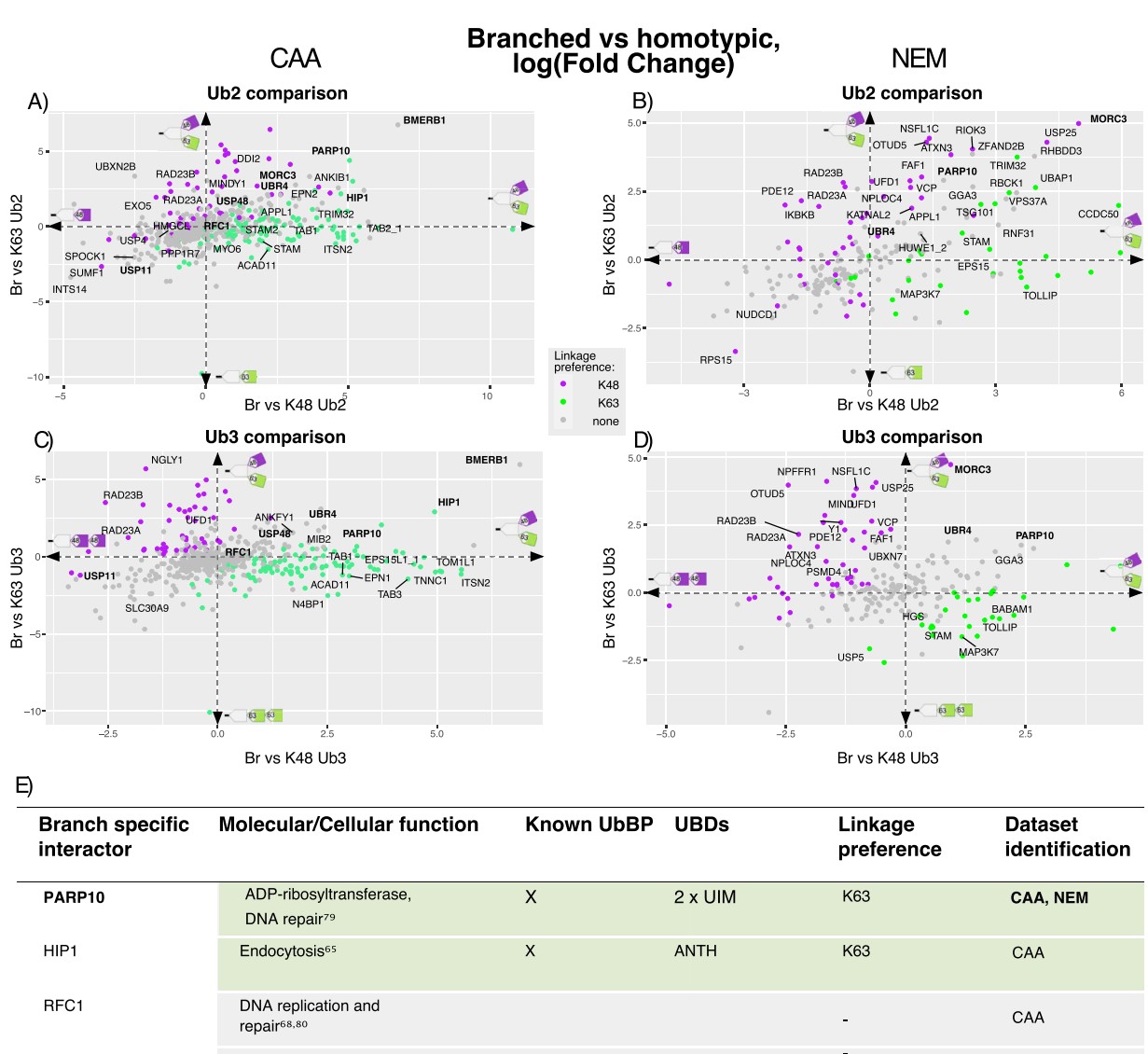

**Figure 4. K48/K63-linked branched Ub chain interactors.**
**(A, B, C, D)** Scatterplots of Br Ub3 versus homotypic Ub2 (A, B) or Ub3 (C, D) interactors. **(A, B, C, D)** Scatterplots (A, C) from the chloroacetamide dataset and (B, D) from the N-ethylmaleimide dataset. Comparison of K63 Ub plotted on the y-axis and K48 Ub plotted on the x-axis. Two-sample moderated $t$ tests on prefiltered Ub-enriched interactors. Dot colour indicates linkage preference by a two-sample moderated $t$ test of K48 Ub3 versus K63 Ub3 (Adj.$P$ < 0.05): purple is significantly enriched on K48, green is significantly enriched on K63, and grey is not statistically significant. Labelled proteins were statistically significant in the Br Ub3 versus homotypic chain comparisons on both axes. Proteins labelled in bold were significant in at least four Br Ub3 versus homotypic chain comparisons across datasets. Adj.$P$ < 0.05. **(C, D, E)** Table of branch-specific interactors identified by significant enrichment on Br Ub3 over both homotypic K48 and K63 Ub3 chains (C, D). PARP10 and UBR4 labelled in bold were significantly enriched on Br Ub3 in every homotypic Ub chain comparison in both datasets. A two-sample moderated $t$ test of prefiltered Ub-enriched interactors, log(FC) > 0, Adj.$P$ < 0.05.

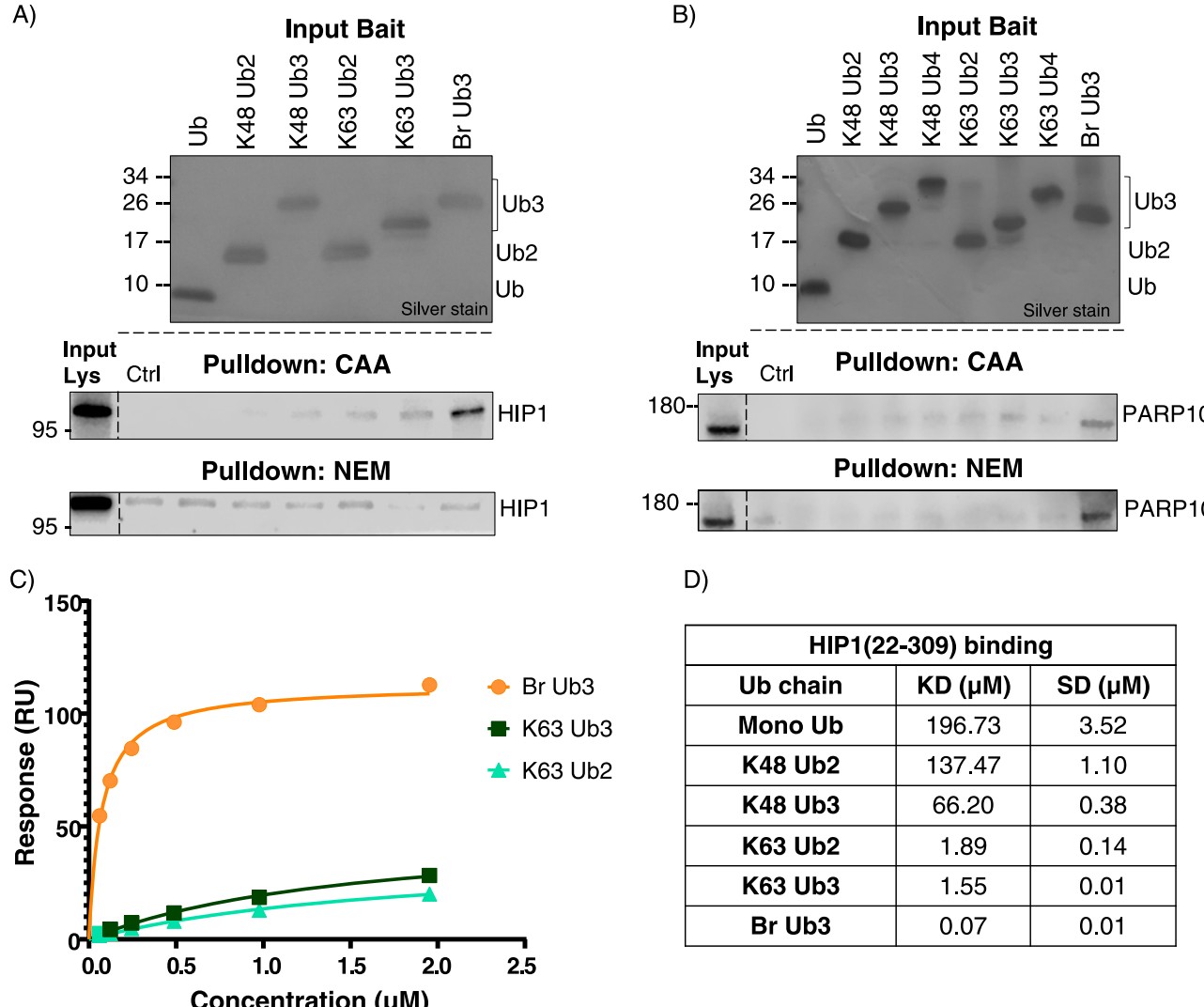

**Figure 5. Further validation of HIP1 and PARP10 branch specificity.**
**(A)** Western blot of Ub pulldown of HIP1 from the chloroacetamide- and N-ethylmaleimide–treated lysate. The same Ub interactor pulldown experiment as in Fig 1B, and thus the same silver stain of input Ub. Pulldown blotted with anti-HIP1 antibody. **(B)** Western blot of Ub pulldown of PARP10 from the chloroacetamide- and N-ethylmaleimide–treated lysate. The same Ub interactor experiment as in Fig S11, and thus the same silver stain of Ub input. Pulldown blotted with anti-PARP10 antibody. **(C)** Overlaid affinity plots of triplicate measurements with determined $K_D$ values and respective SD of K63 Ub2, K63 Ub3, and Br Ub3 chains interacting with immobilized biotinylated HIP1(22-309). Averaged response values (RU) at equilibrium are plotted against the injected concentration ($\mu$M) of respective analytes determined by surface plasmon resonance multicycle format and fitted according to a steady-state affinity model. **(D)** Calculated KD affinities and SDs of HIP1(22-309) with Ub chains measured by surface plasmon resonance and fitted via a steady-state affinity model.

hydroxyurea with a Ubc1 UBA mutant, lacking K48-branching ability (in a Ubc4 KO background) (28). Furthermore, in the aforementioned preprint, Lange et al observed an increase in K48/K63 branched Ub at the site of DNA damage (14 Preprint). Our results also suggest the involvement of K48/K63 branched Ub in Huntington's disease, as we identified the huntingtin-interacting protein HIP1 and UBR4, which has been linked to K11/K48 branched ubiquitination on mutant huntingtin (68), as specific interactors of Br Ub3. Finally, three of our Br Ub3-specific interactors are components of ubiquitination machinery: E3 ligases UBR4 (13, 68) and MIB2 (72), and Cullin–RING ligase substrate adaptor, ANKFY1 (71). The branchpoint may, therefore, be a scaffold for further ubiquitination.

The Ub chain-type specificity of UbBPs in yeast is less well characterized. We provide the first resource of chain-specific Ub interactors in budding yeast. Our data are supported by the identification of known chain linkage–specific UbBPs. We also identified the linkage-type specificity of UbBPs with known UBDs, but previously unknown chain-type preference, including K48-binding Cia1 and K63-binding Duf1 and Gts1. We reproduced Ddi1's preference for longer Ub chains (58) and showed that this was conserved from yeast to humans.

As our screen used native Ub chains, it was essential that we blocked Ub chain disassembly by inhibiting endogenous DUBs. We provide the first comparison of cysteine alkylators NEM and CAA as DUB inhibitors. We observed that lysate treatment with NEM stabilized Ub chains better than treatment with CAA, in line with literature on the potency of NEM

(30). However, NEM also inhibited the Ub binding of some UbBPs, including HIP1, or altered their chain-type specificity, as for IKBKG/NEMO. Notably, in both HIP1 and IKBKG/NEMO, a cysteine residue lies within the UBD, near the site of Ub binding, suggesting that alkylation at this position may be responsible for altered Ub binding (35, 77). These findings bring caution to the use of unspecific alkylating reagents in protein binding studies. For DUB inhibition, more targeted reagents are available, including inhibitors against specific DUBs (87) or broad-spectrum DUB inhibitors such as PR-619 (88). Nonetheless, it is worth noting that PR-619 exhibits reduced efficacy against non-cysteine protease DUBs (89).

In summary, this resource provides readers, interested in the Ub code, with the chain type, linkage, length, and homotypic versus branched preference for K48 and K63 Ub interactors in humans and yeast. Our method cannot discriminate which of these interactors directly binds Ub. To note, indirect Ub binders are also of interest as they may be part of Ub-binding complexes or signalling pathways affected by chain type–specific Ub signals. However, our hits do include proteins with known UBDs and we reproduce the linkage and length preference for many known chain type–specific UbBPs, supporting the reliability of our results. Additional biochemical binding assays are required to establish novel UbBP chain-type specificity using absolute binding affinities, as performed for HIP1. The significant chain-type enriched proteins we identified serve to inform potential models of Ub signalling pathways on which further experiments can be based.

# Materials and Methods

### Cloning

A glycine serine repeat linker containing one cysteine residue was cloned after the ubiquitin (Ub) C-terminus and before a 6x histidine tag in pMD11 (hUb-6H in pETM60) using the Gibson assembly, resulting in the pAW006 plasmid.

### Recombinant protein purification

BL21 Rosetta cells were transformed with expression plasmids and grown overnight at 37°C. At an OD600 of 0.8–1.0, the culture was cooled to 18°C and expression was induced with 0.5 mM IPTG and incubated overnight. Cells were pelleted and resuspended in lysis buffer: 50 mM Tris–HCl, pH 7.5, 150 mM NaCl, 1 mM PMSF. Cells were then lysed with an Avestin EmulsiFlex-C5 homogenizer and cleared of cell debris by centrifugation at 20,000*g* at 4°C for 20 min.

For purification of 6xHis-tagged Ubs, 5 mM imidazole and 3 mM 2-mercaptoethanol were included in the lysis buffer. The lysate was incubated with TALON metal affinity resin (Takara) (3 ml slurry per L culture) for 1 h at 4°C on a rotor. The resin was washed in a gravity column (Bio-Rad) with 4 x wash buffer: 50 mM Tris–HCl, pH 7.5, 150 mM NaCl, 3 mM 2-mercaptoethanol, 10 mM imidazole. Protein was eluted with elution buffer: 50 mM Tris–HCl, pH 7.5, 150 mM NaCl, 300 mM imidazole. A final concentration of 2 mM DTT was added.

For purification of GST-tagged proteins, 4 mM DTT was added to the cleared lysate, which was then incubated with glutathione Sepharose 4B beads (Cytiva—formerly GE Healthcare) (5 ml per litre culture) for 1 h at 4°C. The resin was washed in a gravity column with 4 x wash buffer: 50 mM Tris–HCl, pH 7.5, 150 mM NaCl, 4 mM DTT. Protein was then eluted with elution buffer: 50 mM Tris–HCl, pH 7.5, 150 mM NaCl, 20 mM glutathione.

Purification of untagged Ub was performed by acidic preparation (93). 70% perchloric acid was added dropwise under a fume hood to the cleared lysate until the solution becomes very cloudy. The precipitant was removed by centrifugation at 20,000*g* at 4°C for 20 min. The supernatant was neutralized to pH 7.5 using 10 M NaOH.

All proteins were further purified by size-exclusion chromatography. Protein solutions were concentrated using an Amicon Ultra Centrifugal filter (Millipore) for application to FPLC AKTA pure (GE Healthcare). For Ub purification, a HiLoad 26/600 Superdex 75 pg column (Cytiva) was used. For larger proteins, a Superdex 75 Increase 10/300 GL column (Cytiva) was used. Proteins were

**Plasmids and reagents.**

| Insert | Vector | Source | Identifier |
|---|---|---|---|
| hUb | pETM60 | Maximilian von Delbrück/von Delbrück et al (90) | pMD10 |
| Ub K48R K63R | pETM60 | Lukas Pluska, Sommer Lab | pLP109 |
| Ub-6H | pETM60 | Maximilian von Delbrück/von Delbrück et al (90) | pMD11 |
| Ub-cys-6H | pETM60 | This study, Anita Waltho, Sommer Lab | pAW006 |
| T7Tag_6His_Ub | pET-28a | Klevit Lab, Seattle | pV002 |
| 6H-Ube1 | pET-21 | Klevit Lab, Seattle/Berndsen et al (91) | pV005 |
| GST-CDC34 | pGEX-6p1 | Dötsch Lab, Frankfurt | pMD26 |
| GST-Ubc13 | pGEX-4T1 | Glickman Lab, Haifa/Mansour et al (92) | PMD28 |
| GST-Uev1a | pGEX-6p1 | Maximilian von Delbrück/Mansour et al (92) | pMD29 |
| GST-AMSH | pGEX-6p1 | Lukas Pluska, Sommer Lab, Pluska et al (28) | pLP125 |
| GST-OTUB1 | pGEX-6p1 | Robert-William Welke, Sommer Lab | pRW101 |
| His10-TEV-Avi-tag | pET-15b | Mahil Lambert, Dötsch Lab | pET-15b-His10 -TEV-Avi-tag |
| HIP1 ANTH domain (22-309) | pET-15b-His10 -TEV-Avi-tag | This study, Mahil Lambert, Dötsch Lab | ML117 |

checked by SDS–PAGE and Coomassie staining. Protein concentration was determined using a DC protein assay (Bio-Rad).

### Active UBE1 purification

A Ub-affinity-gel column was prepared to purify active bound UBE1. All steps were carried out at 4°C, unless stated otherwise. 5 ml Affi-Gel 19 slurry (Bio-Rad) was added to a gravity column (Bio-Rad) and washed 3 × with 10 ml $H_2O$. 6 ml of 50 mg/ml 6His-T7tag-Ub purified in MOPS buffer was added and incubated overnight at 4°C on a roller. The supernatant was removed, and then, the unconjugated resin was blocked by incubation with 0.5 ml 1 M ethanolamine, pH 8, for 1 h on a roller. The column was washed 6 × with 0.1 M 3-(N-morpholino)propanesulphonic acid (MOPS), pH 7.2, and stored at 4°C in 0.1 M MOPs, 2% sodium azide, pH 7.2, until use.

UBE1 was expressed as in 4.2.2.1 and then lysed in 50 mM Tris–HCl, pH 8, 0.2 mM DTT, 10 mM $MgCl_2$, 1 mM PMSF with an Avestin EmulsiFlex-C5 homogenizer. The lysate was cleared of cell debris by centrifugation at 20,000$g$ at 4°C for 20 min and then further filtered through a 0.45-$\mu$m filter. Storage buffer was removed from the Ub-affinity-gel column, and it was equilibrated in 50 mM Tris–HCl, pH 8. The filtered UBE1 lysate was added to the prepared Ub-affinity-gel column. The column was topped up with 5 ml ATP and $MgCl_2$, pH 7, to final concentrations of 40 mM of each. The column mix was incubated at RT for 1 h to conjugate the UBE1. The flow-through was removed, and the column was washed ~20 x with 50 mM Tris–HCl, pH 8, 0.5 M KCl (until no contaminants at 280 nm by NanoDrop). 5 ml elution buffer (50 mM Tris–HCl, pH 8, 10 mM DTT) was added and incubated for 10 min before collecting flow-through. This step was repeated six times.

The pooled eluate was dialysed into 50 mM Tris–HCl, pH 8, 150 mM NaCl, 1 mM DTT overnight, using two dialysis steps with sealed dialysis tubing and 4 litre buffer. The final sample was concentrated with an Amicon Ultra-15 filter (MWCO 50).

The Ub-affinity-gel was washed and stored in 0.1 M MOPs with 2% sodium azide. It was regenerated for further use by washing 3 × with 50 mM Tris–HCl, pH 9, 1 M KCl.

### Preparation of Ub chains

Ub chains in which the proximal Ub contains a C-terminal cysteine followed by a 6xHis tag were assembled in vitro using recombinant Ub ligases. For homotypic chain synthesis, the reaction mix contained E1 ligase 2 $\mu$M hUbe1, E2 ligases 25 $\mu$M Cdc34 or 6 $\mu$M Uev1a and 6 $\mu$M Ubc13 for K48 or K63 Ub synthesis, respectively, 1.2 mM Ub, and 0.8 mM 6xHis-cys-Ub. K48/K63 branched chains were synthesized with 2 $\mu$m hUbe1, 10 $\mu$M Ubc1, 8 $\mu$M Uev1a, 8 $\mu$M Ubc13, 0.5 mM 6xHis-cys-Ub, and 1 mM K48R, K63R Ub. Synthesis reactions were performed in 50 mM Tris–HCl, pH 8, 9 mM MgCl2, 15 mM ATP, 5 mM beta-mercaptoethanol (BME) with a total volume of 3–5 ml. Reactions were incubated overnight at 37°C. The next day, the reaction was diluted with 50 mM Tris, pH 7.5, 150 mM NaCl, 7.5 mM imidazole, 1.5 mM BME. 6xHis-tagged chains were then purified with TALON resin (5 ml per 1 ml reaction) and washed 3 x with wash buffer: 50 mM Tris–HCl, pH 7.5, 150 mM NaCl, 3.5 mM BME, 5 mM imidazole. After elution with 300 mM imidazole, Ub chains of

different lengths were separated by size exclusion on FPLC AKTA pure (GE Healthcare) using a HiLoad 26/600 Superdex 75 pg column (Cytiva), with a low flow rate (0.3–0.5 ml/min).

For future immobilization of Ub chains, biotin was conjugated onto the proximal Ub of the chain via maleimide–cysteine reaction. 0.5–1.5 mg Ub chain was reduced by incubation for 1 h at 37°C with 10 x molar excess of tris(2-carboxyethyl)phosphine (TCEP). Reducing agents were then removed by filtering through Pierce Dye and Biotin Removal Column (Thermo Fisher Scientific). The resulting chains were then incubated overnight at room temperature with 10x molar excess of EZ-Link Maleimide-PEG2-Biotin (Thermo Fisher Scientific). The next day, excess biotin was quenched with a final concentration of 10 mM DTT. Excess biotin and DTT were removed by 6 x sequential dilution and concentration using a 3 KD Amicon Ultra Centrifugal filter (Millipore). The yield of biotin conjugation was tested by intact MS with an Agilent 6230 b LC-MS/TOF mass spectrometer. Chain concentration was determined by a DC protein assay (Bio-Rad).

### Cell culture/lysate preparation

For Ub interactor copulldown, wild-type *S. cerevisiae*, yeast, and HeLa cell lysate were prepared.

5 litre yeast culture was grown to 1 OD600, harvested at 4,000$g$ for 5 min, washed in 25 ml water and 1 mM PMSF, and frozen at –80°C until further use. The cell pellet was resuspended in yeast lysis buffer (200 $\mu$l per 100 ml culture): 50 mM Tris–HCl, pH 8, 150 mM NaCl, 0.4% NP-40 (ca630; IGEPAL), 1 mM PMSF, 5% glycerol, and aliquoted into 2-ml Eppendorf tubes for lysis. Glass beads (Carl Roth) (≈200 $\mu$l per tube) were added, and cells were lysed on with a vortex at max. speed for 5 × 1 min (incubation on ice in between vortexing). Minimal lysis buffer: 50 mM Tris, pH 8, 150 mM NaCl, 0.1% NP-40, 10 mM CAA, 1 mM PMSF (300 $\mu$l per tube), was added. Cell debris was removed by centrifugation at 1,000$g$ for 3 min at 4°C. The supernatant was transferred to a fresh 1.5-ml Eppendorf tube, and the lysate was cleared by centrifugation at 20,000$g$ for 5 min at 4°C. Pierce BCA assay (Thermo Fisher Scientific) was used to measure the protein concentration of the lysate.

HeLa cells were grown in DMEM, supplemented with 10% FBS and 1% penicillin/streptomycin at 37°C, 5% $CO_2$, and 90% humidity. 50 × 15 cm dish HeLa cells were harvested with trypsin (3 ml per 15 cm dish) at 90% confluency. Cell pellets were washed 4 x with PBS (Sigma-Aldrich) and frozen at –80°C for further use. Cells were resuspended in HeLa lysis buffer (450 $\mu$l per 15 cm dish): 150 mM NaCl, 50 mM Tris–HCl, pH 8.0, 0.5% (vol/vol) IGEPAL, 5% (vol/vol) glycerol, 1 mM PMSF, 1x protease inhibitor mix, briefly vortexed, and incubated for 45 min at 4°C on a rotor. The lysate was centrifuged at 14,000$g$ for 15 min at 4°C. Pierce BCA assay (Thermo Fisher Scientific) was used to measure the protein concentration of the lysate. Before use in the Ub interactor pulldown, 1 mM NEM or 10 mM CAA was added as a DUB inhibitor and incubated with the lysate for 1 h.

### Ub interactor pulldown for mass spectrometry

Ub interactor pulldown was done in quadruplicate for each chain type or resin-only control. 25–50 $\mu$g Ub chains were immobilized on

 **Life Science Alliance**

streptavidin magnetic Sepharose resin (Cytiva) by incubation on a rotor for 1 h at 4°C in 200 $\mu$l binding buffer (50 mM Tris–HCl, pH 8.0, 150 mM NaCl, 0.1% NP-40). Immobilized chains were washed once with binding buffer. 2.5–4 mg HeLa or yeast lysate was added to immobilized chains and incubated overnight on a rotor at 4°C. The next day, resin was washed 3 x with 1 ml wash buffer (50 mM Tris–HCl, pH 8, 150 mM NaCl). The enriched material was eluted by an on-bead digest, as detailed below.

### Sample preparation and mass spectrometry

Sample preparation was conducted using on-bead tryptic digestion, adhering to the protocol established (94). Briefly, washed beads were incubated in digestion buffer: 2 M urea, 50 mM Tris (pH 7), 1 mM dithiothreitol (DTT), and 0.4 $\mu$g sequencing-grade trypsin (Promega), for 1 h at 25°C with continuous agitation on a Thermomixer compact shaker (Eppendorf) operating at 1,000 rpm. After the incubation period, the supernatant was carefully transferred to a fresh tube. The beads were washed again 2 x with urea/Tris buffer and each time combined with the supernatant with previous steps. Proteins were reduced with 4 mM DTT for 30 min and alkylated using 10 mM CAA for 45 min at 25°C whilst shaking at 1,000 rpm on a Thermomixer compact shaker (Eppendorf). Proteins were subjected to a second digest overnight with 0.5 $\mu$g trypsin and incubated at 25°C whilst shaking at 700 rpm on a Thermomixer compact shaker (Eppendorf). After the tryptic digestion, we employed stage-tips, following the procedure described (95) to remove salts and impurities from the samples. For human CAA-treated samples, peptides were cleaned up using a peptide-based SP3 approach (96). Briefly, peptides in aqueous solution were incubated with SP3 beads at a ratio of 200:1 beads:protein. ACN was added to a final concentration of 95%, and beads were subsequently washed 3 x with 100% ACN. Peptides were eluted from beads in 50 $\mu$l LC-MS–grade water.

The samples were then subjected to liquid chromatography–mass spectrometry (LC-MS) measurements using an Orbitrap Exploris 480 mass spectrometer (Thermo Fisher Scientific) in conjunction with an EASY-nLC 1200 system (Thermo Fisher Scientific). The mass spectrometer was operated in a data-dependent mode, and a 110-min gradient was applied.

### Proteomics analysis

For the analysis of mass spectrometry data, we used MaxQuant version 2.0.3.0 (97) incorporating MaxLFQ-based quantitation (98) and enabling the match-between-runs algorithm. Carbamidomethylation of cysteine residues, acetylated protein N-termini, and oxidized methionine were designated as variable modifications. Peptides containing cysteine residues were not used in quantitation. The Andromeda search was performed using a UniProt human or yeast database from 2022 including protein isoforms, along with a list of common contaminants. An FDR cut-off of 0.01 was applied on the PSM and protein level.

Subsequent to data acquisition and initial processing, downstream data analysis was conducted in the R programming environment (v4.2.1) using iBAQ values for quantitation.

In R (v4.2.1), data were reverse-filtered, dropout samples were removed, and IBAQ values were used to filter by valid values (≥3 per chain type), median-normalized, and imputed using a normal distribution with the downshift approach. For use in further analysis, Ub-enriched proteins were prefiltered by significance in at least one Ub pulldown in a two-sample moderated $t$ test against the control bead–only pulldown (adjusted $P$-value < 0.05, log (fold change) < 0). This filter was then applied to data at the step before normalization and imputation. Subsequently, filtered data were median-normalized and imputed using a normal distribution with downshift. Data that have undergone this prefiltering are described as prefiltered Ub-enriched proteins in the text and figure legends.

Lists of expected UbBPs in humans and budding yeast were created from collating proteins from the Gene Ontology (GO) terms Ub-binding GO:0043130 and linear Ub-binding GO:1990450, UBD-containing proteins from the integrated annotations for Ubiquitin and Ubiquitin-like Conjugation Database, ver 2.0 (iUUCD) (38) (http://iuucd.biocuckoo.org/), and more recently discovered UBD-containing proteins from cited literature.

Fisher's exact test was performed to compare the proportion of expected UbBPs in the unfiltered and filtered datasets in R. All other statistical tests were performed in R using the Shiny app ProTIGY provided by the Broad Institute on GitHub (https://github.com/broadinstitute/protigy). A moderated F test was used to compare all pulldown samples. A two-sample moderated $t$ test was used for pairwise comparisons. To identify significant hits, a significance threshold of < 0.05 was applied for Benjamini–Hochberg corrected adjusted $P$-values (Adj.$P$), unless otherwise stated. Data visualization was performed in R using the tidyr, dplyr, tibble, tidyverse, pheatmap, ggplot2, stringr, corrplot, and VennDiagram packages. Metascape was used for Gene Ontology (GO) enrichment (39) with express analysis settings: minimum overlap of 2, $P$ < 0.01, and minimum enrichment of 1.5.

### Ub interactor pulldown validation for Western blot

14 $\mu$g Ub chains were immobilized on streptavidin magnetic Sepharose resin (Cytiva) by incubation on a rotor for 1 h at 4°C in 100 $\mu$l binding buffer (50 mM Tris–HCl, pH 8.0, 150 mM NaCl, 0.1% NP-40). Immobilized chains were washed once with binding buffer. 1 mg HeLa or yeast lysate was added to immobilized chains and incubated overnight on a rotor at 4°C. The next day, resin was washed 3 × with 0.5 ml wash buffer (50 mM Tris, pH 8, 150 mM NaCl). The pulled-down material was eluted in 25 $\mu$l SDS sample buffer with 2-mercaptoethanol and boiled at 95°C for 10 min 10 $\mu$l samples were run on SDS–PAGE and Western blot against Ub and proteins of interest.

### Western blot

SDS–PAGE was done using homemade Bis-acrylamide gels or Mini-PROTEAN TGX gradient gels (Bio-Rad). Gels were blotted onto PVDF membranes. Membranes were blocked in 5% milk powder or 10% Roti-Block (Roth). Primary antibodies were added overnight at 4°C in 5% milk powder or 10% Roti-Block (Roth)—1:1,000 Ub antibody P4D4 (Santa Cruz Biotechnology), 1:3,000 Hip1 (22231-1AP;

Proteintech), 1:000 PARP10/ARD10 (NB100-2157; Novus Biologicals), 1:1,000 Rad23b (E-AB-62188; Elabscience), 1:1,000 Dsk2 (ab4119-100; Abcam), 1:4,000 Rad23 (Sommer Lab), 1:1,000 Epn2 (Invitrogen), DDI2 (AB197081; Abcam), 1:1,000 DDI1 serum (Jeffrey Gerst, Weizmann Institute of Science (99)), 1:1,000 APPL1 (D83H4; Cell Signaling), 1:1,000 CCDC50 (AB127169; Abcam), 1:1,000 FAF1 (1027-1-AP; Proteintech), 1:1,000 ZFAND2B (Sigma-Aldrich), 1:1,000 Riok3 (13593-1-AP; Proteintech), 1:1,000 USP11 (22340-1-AP; Proteintech), 1:5,000 CDC48 (Sommer Lab), 1:1,000 Vps9 (Scott Emr, Cornell University (100)), 1:1000 YUH1 serum (Tingting Yao, Colorado State (101)). Anti-mouse IgG HRP (Sigma-Aldrich) and anti-rabbit IgG HRP (Sigma-Aldrich) were used at 1:10,000 as secondary antibodies. Immuno-blots were visualized using an Odyssey XF imager (LI-COR).

### Ub chain disassembly (UbiCRest)

1.25 μg ub chains were incubated with 1 μM DUB, either K48-specific OTUB1 or K63-specific AMSH, in disassembly buffer (50 mM Tris–HCl, pH 7.5, 50 mM NaCl, 10 mM DTT) for 45 min at 37°C. Reactions were stopped with SDS–DTT sample buffer. Samples were run on SDS–PAGE and Western blot with anti-Ub antibody and imaged using an Odyssey XF imager (LI-COR).

### Biotinylated HIP1 (22-309) preparation

DNA coding for amino acids 22-309 (ANTH domain) of HIP1 codon-optimized for *E. coli* expression was bought from Eurofins Genomics. The coding sequence was cloned into a pET-15b-His10-TEV-Avi-tag vector. The plasmid was transformed into *E. coli* BL21(DE3) cells and grown in 2xYT medium. Protein expression was induced with 1 mM IPTG after reaching an OD of 0.6, and expression was carried out for 16 h at 18°C. Cells were harvested by centrifugation, resuspended in buffer A (50 mM Bis-Tris, pH 6.5, 0.4 M NaCl, 20 mM β-mercaptoethanol, 20 mM imidazole) supplemented with RNase (Sigma-Aldrich), DNase (Sigma-Aldrich), and protease inhibitor cocktail (homemade), and lysed by soni-cation. The lysate was cleared by centrifugation at 4°C and applied onto a pre-equilibrated immobilized metal affinity chromatography (IMAC) column (HiTrap IMAC Sepharose FF, Cytiva). The bound protein was washed with buffer A and eluted by a step gradient with buffer A supplemented with 500 mM imid-azole. The eluted protein was dialysed concurrently into buffer A and subjected to digestion with self-made TEV protease. Sub-sequently, TEV protease and undigested protein were separated using a reverse IMAC step. Purified proteins were finally subjected to size-exclusion chromatography (SEC, HiLoad 16/600 Superdex 75; Cytiva) in SEC buffer (50 mM Hepes, pH 7.5, 150 mM NaCl, 0.5 mM TCEP).

HIP1 (22-309) with an Avi-tag underwent enzymatic biotinylation in vitro with the *E. coli* biotin ligase BirA (expressed as a fusion protein with GFP) in a 1:50 molecular ratio in SEC buffer, along with supplementation of 10 mM ATP, 10 mM MgCl$_2$, and 0.5 mM biotin for 16 h at 16°C. Subsequently, the reaction mixture was subjected to a Superdex 75 10/300 column (Cytiva) for separation and for buffer exchange to SPR buffer. Finally, the biotinylated HIP1 (22-309) was subjected to analysis by LC-ESI-TOF mass spectrometry.

### SPR

SPR experiments were performed on Biacore T200 at 25°C using 50 mM Hepes, pH 7.5, 300 mM NaCl, 0.5 mM TCEP, 0.05% Tween-20, and 1% BSA as the running buffer. Series S Sensor Chips NA (Cytiva) were used according to the manufacturer's recommendations: 3x injection of 50 mM NaOH, 1 M NaCl for 60 s to condition the surfaces with a subsequent injection of recombinant biotinylated HIP1 protein at 2 μg/ml diluted in running buffer. A final response of 900 RU was reached for three flow channels by immobilizing the protein for 90 s at 10 μl/min. One flow channel was used as an empty reference surface without protein injection.

Each ubiquitin (Ub) analyte was measured at a flow rate of 30 μl/min using multicycle conditions injecting single concentrations during each cycle increasing from 60 nM up to 250 μM over the reference and protein surfaces. Resulting sensorgrams were ref-erenced, blank-subtracted, and evaluated.

Evaluation was carried out using the steady-state affinity model:

$$Req = \frac{C \times Rmax}{KD + C}$$

where C is the injected concentration, Rmax is the maximal re-sponse, and Req is the response at the steady state to determine the respective $K_D$ values.

## Data Availability

The mass spectrometry proteomics data have been deposited to the ProteomeXchange Consortium (http://proteomecentral.proteomexchange.org) via the PRIDE partner repository (102) with the dataset identifier PXD051890.

## Supplementary Information

## Acknowledgements

We would like to thank some of our colleagues at the Max Delbrück Centre for Molecular Medicine: Ernst Jarosch for his advice during the project and article proofreading, Mandy Gerlach for her technical assistance with Western blotting, Robert-William Welke for providing OTUB1 and AMSH DUBs, Sabine Meyer and Anja Schutz from the protein production facility for their technical assistance with the LC-MS/TOF mass spectrometer, and Mohamad Haji from the proteomics facility for his technical assistance with the LC-MS instrument. In addition, thanks to those who kindly gifted us antibodies. We would like to acknowledge our funding agencies: the joint DFG grant DO 545/17-1 | SO 271/9-1, and the Structural Genomics Consortium (SGC), a registered charity (No:1097737) that received funds from Bayer AG, Boehringer Ingel-heim, Bristol Myers Squibb, Genentech, Genome Canada through Ontario Genomics Institute, Janssen, Merck KGaA, Pfizer, and Takeda. This project received funding from the Innovative Medicines Initiative 2 Joint Undertaking (JU) under grant agreement No. 875510. The JU receives support from the European Union's Horizon 2020 research and innovation programme, EFPIA, Ontario Institute for Cancer Research, Royal Institution for the Advancement

of Learning/McGill University, Kungliga Tekniska Hoegskolan, and Diamond Light Source Limited. Disclaimer: This communication reflects the views of the authors, and JU is not liable for any use that may be made of the information contained herein. This work was supported by the Deutsche Forschungsgemeinschaft (DFG—German Research Foundation) under grant agreement SFB 1470 "HFpEF" (Project B05) and SFB 1588 "Neuroblastoma Evolution" (Project A06) to P Mertins.

## Author Contributions

A Waltho: conceptualization, data curation, formal analysis, validation, investigation, visualization, methodology, and writing—original draft.
O Popp: software, formal analysis, and investigation.
C Lenz: validation and investigation.
L Pluska: conceptualization and supervision.
M Lambert: validation.
V Dötsch: conceptualization and supervision.
P Mertins: conceptualization, supervision, and methodology.
T Sommer: conceptualization, supervision, and funding acquisition.

## Conflict of Interest Statement

The authors declare that they have no conflict of interest.

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
