## [Reviewer comments · Life Science Alliance]

Life Science Alliance

K48- and K63-linked ubiquitin chain interactome reveals branch- and length-specific interactors.

Anita Waltho, Oliver Popp, Christopher Lenz, Lukas Pluska, Mahil Lambert, Volker Dötsch, Philipp Mertins, and Thomas Sommer

DOI: <https://doi.org/10.26508/lsa.202402740>

Corresponding author(s): Anita Waltho, Max Delbrück Center for Molecular Medicine and Thomas Sommer, Max Delbrück Center for Molecular Medicine

Review Timeline:

Submission Date:	2024-03-27
Editorial Decision:	2024-03-28
Revision Received:	2024-05-08
Accepted:	2024-05-08

Transaction Report:

Please note that the manuscript was previously reviewed at another journal and the reports were taken into account in the decision-making process at *Life Science Alliance*.

Reviews

Referee #1

Report for Author:

Waltho et al carry out proteomic analysis to identify proteins that bind to different lengths and linkage types of ubiquitin. The authors use different lengths of K48 and K63 chains of ubiquitin and a ubiquitin trimer that contains both a K48 and K63 linkage (model for a branched chain). Binding proteins are captured on beads, eluted and analysed by mass spectrometry. Many known ubiquitin interactors are identified as are many proteins that had not previously been shown to be ubiquitin binders. The proteomic analysis is nicely done, but the paper comes across as rather preliminary. In its present form the paper doesn't quite make it as a resource, and it doesn't contain enough biology to make it as a full paper.

The difficulty in viewing this paper as a resource is that interaction studies, such as those used in this study, have the limitation that not all proteins enriched significantly in a 'bait' condition, are direct binders. Therefore, the reader cannot be sure if a protein specifically enriched by one of their ubiquitin topologies is itself a ubiquitin binding protein, or if it is pulled down indirectly by virtue of it interacting with a direct ubiquitin binder. This critically weakens the argument for publication as a resource as many of the proteins enriched in a specific condition will not actually be ubiquitin binders. The authors could carry out additional bioinformatic analysis to search their entire list for all known ubiquitin-binding sequences and domains as they did for a small number of proteins indicated in Figs. 3G, 4E. Even if the sequence homology is low, it would also be possible to use AlphaFold to look for known structures of Ubds, UIMs etc in their shortlist of proteins. They may also consider searching those proteins with no apparent ubiquitin-binding domains for common sequence features, which may reveal a novel ubiquitin-binding domain. Thus, a summary spread sheet with the proteomics information and the presence of known ubiquitin-binding elements would lend weight to the argument that individual identifications may be directly binding ubiquitin. It is not clear what proportion of the total contains known ubiquitin binding domains. If the proteins without known domains are in the majority then it would be difficult to make the case for the paper being viewed as a resource.

A couple of proteins were followed up to show that binding of a unique ubiquitin topology was direct. The result with HIP1 was very interesting as it demonstrated specific binding to a branched chain. Clearly this is the starting point for a study demonstrating the biological relevance of such an interaction. What domain in the protein is responsible for this interaction ? and if this domain is mutated what are the biological consequences ? If such studies were carried out then the paper would fall nicely into the full paper category.

The comparisons between NEM and chloroacetamide for the effect of protein alkylation on protein binding were quite surprising and very useful to bear in mind for studies such as these. This is information that would be appreciated by the proteomics community.

Referee #2

Report for Author:

With the recognition that functionally important polyubiquitin (polyUb) signals can be comprised of branched heterotypic polyUb chains, there is a need to identify the cellular proteins used to recognize them. This study by Waltho et al. aimed to provide researchers with a database of (poly)Ub interacting proteins together with information about their relative specificities for monoUb, K48 or K63-linked homotypic polyUb (n= 2 or 3), and K48/K63-branched triUb. The results were obtained primarily from pulldown/LC-MS proteomics experiments that used synthetic polyUb chains to capture proteins from HeLa or yeast cell lysates; additionally, the authors found that different proteins were identified depending on which alkylating agent (i.e., NEM or CAA, chloroacetamide) was used to inactivate deubiquitinating enzymes in the lysates. Unfortunately, the limited scope of the study and high variability in the results severely compromise the reliability and utility of the work. These and some other issues are elaborated below.

Results from replicate pulldowns were presented (n = 3 or 4; see Fig. 2C,D) to identify ubiquitin binding proteins using various conditions and forms of polyUb as bait. The heat map data show that, for the same condition, many proteins showed

highly variable abundance (or ranking). For example, individual proteins identified in Cluster 4 as branched K48/K63 interactors (Fig. 2C,D) returned scores ranging from -2 to +2. Given this high variability and no or very limited validation of the results by western blot or orthogonal binding assays, the results do not appear to be sufficiently robust for use as a resource.

Differences observed when either NEM or CAA was used to inactivate DUB activities in lysates feature prominently in the paper, but the source (and reliability) of those differences is unclear. Whereas the authors attribute the differences to modification of (poly)Ub interactors at the ubiquitin-binding interface, that was never established; rather, different extents of polyUb hydrolysis remain a plausible explanation. The authors contend - but do not show - that the immobilized polyUb bait exceeds the binding proteins in the lysates. In principle, endogenous DUBs in the lysates could cleave Ub from the biotinylated peptide tail that tethers it to the resin. Also, even limited hydrolysis of the branched K48/K63 polyUb could specifically enrich for either K63 or K48 diUb and bias the pulldown results. The authors note (p. 4) for Fig. 2 that among the many dozens of proteins in Cluster 1, only 4 appear in both the NEM and CAA datasets; to me, differential modifications of the ubiquitin-binding interfaces seems very unlikely as the explanation. In light of the emphasis given to the differences in the results with NEM vs CAA, they need further exploration. HIP1 was recovered in the branched K48/K63 polyUb pulldown treated with CAA but not NEM. As HIP1 binding was assayed in vitro with recombinant proteins (Fig. 5), the authors could test if treatment of recombinant HIP1 with either reagent affects the binding.

The recent paper by Lange et al. (ref. 14) also identifies binders that prefer branched K48/K63 polyUb. Waltho et al. indicate a few cases where their results agree, but more often the results disagree; why is not discussed.

Finally, many aspects of the manuscript need clarification and other improvements. Important experimental details are lacking (e.g., how the recombinant HIP1 was made; what is the sequence of the "C-terminal" Cys-containing Ub protein). UBP is generally used to indicate a large subset of yeast DUBs; the use of "UBP" by Waltho et al. to signify ubiquitin binding protein is confusing. Most of the figures need more detailed legends. Ubiquitin has a C-terminal carboxylate, not "hydroxide" (p. 1).

Referee #3

Report for Author:

Ubiquitylation is an important posttranslational modification that can determine the fate of modified proteins based on the linkage type between ubiquitin molecules in polymeric conjugates. The specificity of this ubiquitin code remains incompletely understood, as differences between chain types are sometimes quantitative in nature. Moreover, the consequences of ubiquitylation can also be determined by the number of ubiquitin molecules incorporated into a polymer, but how chain length is read out in cells, and when this is important, is not really known. Studies on further understanding how ubiquitin conjugates are read out and translated into biological reactions are therefore important.

In this manuscript, the authors address these questions by dipping ubiquitin dimers or trimers into cell extracts and determining bound proteins by mass spectrometry. This is not necessarily a novel approach, and previous investigations in this direction resulted in limited new knowledge about linkage-specific ubiquitin signaling. I am afraid the same is true for this study. For most of the study, the current approach does not discriminate between direct or indirect binders, a question that is important because a protein might be part of a complex that contains multiple ubiquitin recognition modules with distinct linkage specificity; such proteins might imply apparent preferences for a particular chain type (for example branched chains), while in reality their binding behavior reflects a combination of multiple independent binding events of the complex to different components of the conjugate. Moreover, lysates take proteins out of their natural context, i.e. a protein with apparent specificity for a particular linkage type or length might never encounter such chains if the respective enzymes and effectors do not localize to the same sites. Thus, at least some follow-up in cells is needed to show that anything documented here occurs without disrupting cell integrity and to show that any such interaction would be important. Third, whether a small preference seen at one concentration of a particular chain type is meaningful in cells depends on local concentrations of the modified protein and its effector; without knowing K_d values, concentrations of both partners in cells, and potential complex formation (i.e. local concentrations), the subtle preferences documented here might be an overestimate of the effects of linkage type or chain length. Finally, the vast majority of conjugates contain more than two or three moieties, but

if principles uncovered for the minimal building blocks investigated here are important in the natural context of a more complex conjugate is not known. Given these significant issues, the current study appears to be rather preliminary.

Additional points:

1. I did not find information where exactly the Cys residue was introduced "near the C-terminus of ubiquitin". If this includes residue 74, which is often recognized by ubiquitin-binding proteins, it might inactivate an important site that contributes to chain length-dependent binding. This would complicate the interpretation of these results. Please provide this information in the main text of the paper. If Arg74 was used, please also repeat at least some experiments with an internal Cys mutant of ubiquitin that is distant from the I44 patch to ensure that adding biotin to this site does not interfere with recognition (alternatively, add some small protein and attach biotin to this protein).

2. Fig. 3E/F needs inputs and bait-immobilized ubiquitin. This experiment provides little support for chain length- or linkage-specificity, as it is unclear how much ubiquitin was immobilized and how much protein was present in lysate.

3. It is generally disappointing how little follow-up was done from the mass spec. To really state that a protein has a preference for a certain linkage or chain length, quantitative binding experiments (for example by either fluorescence polarization, FRET, or SPR) would be required. Only Kd values determined by these methods would provide convincing information about binding specificity for a particular chain length or linkage. This is a critical issue, as the authors ascribed specificity based on relatively small differences in z scores that might not be relevant dependent on expression levels in cells, accessibility to a substrate in cells etc. This has been done for HIP1, where the data is nice, but would really have to be repeated much more throughout the paper. Also, some sort of validation in cells would be required. This would likely require much more work, such as identification of the responsible ubiquitin binding domains, mutations in this domain that reduce association with the preferred chain type and then analysis of any effects on ubiquitin binding in cells. I am not sure whether this could realistically be done within the context of a revision.

4. I am worried about their interpretation that branched ubiquitin chains contain information of each chain type. First, they see completely different correlations in each of their experimental systems (yeast vs. HeLa DUBin.1 vs HeLa DUBin.2). This suggests to me that correlations are driven by some idiosyncrasy of the experiment, not the biological system. Second, I am not sure that simply by looking at the branch point ubiquitin anyone should draw conclusions about signaling functions of branched chains, as most branched conjugates contain much longer and many more chain blocks (it is unlikely that branched chains contain a single branch point). Unfortunately, I believe that at the very least, much longer branched chains would need to be investigated, and substrates that are modified with them would also matter.

5. Fig 5B: apparent preference for branched chains mostly reflects differences in how much ubiquitin had been immobilized to the beads.

Minor point:

1. This manuscript would profit from proofreading by a native English speaker. There are several mistakes throughout the manuscript that are rather confusing ("we sort to expand..."). I apologize for pointing this out, but hope these mistakes could be corrected prior to publication.-----

Authors' Response to Reviewers

We thank the reviewers for critical reading of the manuscript and also for their helpful comments. In light of the manuscript rejection at another Journal, we, the authors, have responded to the reviewers' comments and made adjustments to the manuscript for submission at Life Science Alliance. Please find our responses indented within the reviewers' comments.

Referee #1:

Waltho et al carry out proteomic analysis to identify proteins that bind to different lengths and linkage types of ubiquitin. The authors use different lengths of K48 and K63 chains of ubiquitin and a ubiquitin trimer that contains both a K48 and K63 linkage (model for a branched chain). Binding proteins are captured on beads, eluted and analysed by mass spectrometry. Many known ubiquitin interactors are identified as are many proteins that had not previously been shown to be ubiquitin binders. The proteomic analysis is nicely done, but the paper comes across as rather preliminary. In its present form the paper doesn't quite make it as a resource, and it doesn't contain enough biology to make it as a full paper.

The difficulty in viewing this paper as a resource is that interaction studies, such as those used in this study, have the limitation that not all proteins enriched significantly in a 'bait' condition, are direct binders. Therefore, the reader cannot be sure if a protein specifically enriched by one of their ubiquitin topologies is itself a ubiquitin binding protein, or if it is pulled down indirectly by virtue of it interacting with a direct ubiquitin binder. This critically weakens the argument for publication as a resource as many of the proteins enriched in a specific condition will not actually be ubiquitin binders. The authors could carry out additional bioinformatic analysis to search their entire list for all known ubiquitin-binding sequences and domains as they did for a small number of proteins indicated in Figs. 3G, 4E. Even if the sequence homology is low, it would also be possible to use AlphaFold to look for known structures of Ubds, UIMs etc in their shortlist of proteins. They may also consider searching those proteins with no apparent ubiquitin-binding domains for common sequence features, which may reveal a novel ubiquitin-binding domain. Thus, a summary spread sheet with the proteomics information and the presence of known ubiquitin-binding elements would lend weight to the argument that individual identifications may be directly binding ubiquitin. It is not clear what proportion of the total contains known ubiquitin binding domains. If the proteins without known domains are in the majority then it would be difficult to make the case for the paper being viewed as a resource.

Concerns over the pulldown of indirect binders is obviously an important point raised by both reviewer 1 and 3. In response, we added additional clarifying sentences to the discussion to highlight this limitation and also point out the interesting information indirect binders also provide (475-478):

"Our method cannot discriminate which of these interactors directly bind Ub. To note, indirect Ub-binders are also of interest as they may be part of Ub-binding complexes or signalling pathways affected by chain type-specific Ub signals."

Furthermore, in response to reviewer 1's suggestion to analyse the 'entire' protein list for UBDs and create a summary spreadsheet, we compiled a table (Supplementary Table 1) for each experiment with all pre-filtered proteins (proteins enriched on Ub versus control) annotated with the known Ub-binding domains (UBDs) (from IUUCD database and literature search) and Ub-binding Gene Ontology terms, along with the summarized statistical test data. This table makes it easy for readers to search through proteins with known UBDs in the data, look up their proteins of interest or see general enrichment patterns of different proteins. We refer to it in the text (lines 152-155):

"We compiled a summary table of all subsequent statistical comparisons combined with known UBD information for all prefiltered protein isoforms from CAA- and NEM-treated human, and yeast datasets (S.Table 1 s1-3)."

Reviewer 1 additionally suggested to carry out bioinformatic analysis of proteins without known binding domains for sequence or structural homology, in order to find novel UBDs. This is an interesting suggestion and probably helpful for the further analysis of the dataset. Our dataset aims at being a resource for other groups to mine for information such as this, which would no doubt prompt further research and discovery.

In response to reviewer 1's comment that it was not clear what proportion of the total (proteins identified) contain known UBDs, we clarified this in the text with numbers from the new aforementioned summary, alongside the already included (but now updated) proportion of 'expected Ub binders' (defined as either known UBD-containing or belonging to the GO terms Ub-binding 0043130 or linear Ub-binding 1990450). In reviewer 1's opinion the majority of prefiltered proteins should contain known UBDs. This was not the case, however, in order to prove that expected Ub binders are enriched, we calculated the enrichment factors of expected Ub binders in the prefiltered proteins, compared to the whole deep proteome. In comparison to the whole deep proteome from Nagaraj et al¹ expected UBPs were enriched in our prefiltered protein isoform list

by an enrichment factor of 11.68 for CAA and 19.22 for NEM. We added this information to our results section. We also used a Fisher's exact test to show that prefiltering significantly increased the proportion of expected Ub binders compared to all protein isoforms identified. (lines 139-152)

A couple of proteins were followed up to show that binding of a unique ubiquitin topology was direct. The result with HIP1 was very interesting as it demonstrated specific binding to a branched chain. Clearly this is the starting point for a study demonstrating the biological relevance of such an interaction. What domain in the protein is responsible for this interaction? and if this domain is mutated what are the biological consequences? If such studies were carried out then the paper would fall nicely into the full paper category.

Of course, we know that particularly the identification of Huntingtin-interacting protein 1 (HIP1) as a branch-specific binder is a finding of great potential. However, to understand if, why and how the Ub branch signal is involved in Huntington's disease would require a lot of further experimentation, which we will leave to a later and more specific publication. This paper is intended as a resource paper, providing information to other scientists that are interested in that topic.

The comparisons between NEM and chloroacetamide for the effect of protein alkylation on protein binding were quite surprising and very useful to bear in mind for studies such as these. This is information that would be appreciated by the proteomics community.

We thank reviewer 1 and agree that the result was an interesting one which we too believe should influence future proteomic pulldown studies.

Referee #2:

With the recognition that functionally important polyubiquitin (polyUb) signals can be comprised of branched heterotypic polyUb chains, there is a need to identify the cellular proteins used to recognize them. This study by Waltho et al. aimed to provide researchers with a database of (poly)Ub interacting proteins together with information about their relative specificities for monoUb, K48 or K63-linked homotypic polyUb (n= 2 or 3), and K48/K63-branched triUb. The results were obtained primarily from pulldown/LC-MS proteomics experiments that used synthetic polyUb chains to capture proteins from HeLa or yeast cell lysates; additionally, the authors found that different proteins were identified depending on which alkylating agent (i.e., NEM or CAA, chloroacetamide) was used to inactivate deubiquitinating enzymes in the lysates. Unfortunately, the limited scope of the study and high variability in the results severely compromise the reliability and utility of the work. These and some other issues are elaborated below.

Results from replicate pulldowns were presented (n = 3 or 4; see Fig. 2C,D) to identify ubiquitin binding proteins using various conditions and forms of polyUb as bait. The heat map data show that, for the same condition, many proteins showed highly variable abundance (or ranking). For example, individual proteins identified in Cluster 4 as branched K48/K63 interactors (Fig. 2C,D) returned scores ranging from -2 to +2. Given this high variability and no or very limited validation of the results by western blot or orthogonal binding assays, the results do not appear to be sufficiently robust for use as a resource.

Based on our heatmaps, we disagree with reviewer 2's general statement, 'for the same condition, many proteins showed highly variable abundance'. However, we accept that there is some variability in abundance between some of the technical replicates. This is due to the nature of a pulldown experiment, especially in the case of capturing UBPs as these are often transient weak interactions.

We agree with reviewer 2 that immunoblotting is a good way to validate the initial hits. We have done that for a number of proteins from novel enrichment groups - proteins with preference for longer or branched Ub chains. We were happy to see that those tested hits could be confirmed in this secondary test. Of course, a limitation of this analysis was the availability and strength of specific antibodies. We chose not to validate linkage preference by Western Blot (apart from RAD23B and EPN2 as controls in Figure 1B), as many proteins with linkage preference have already been identified and we reproduced these enrichment patterns, supporting the reliability of our data (already written in text, lines 193-197, 207-210, 241-243, 245-248).

We as well agree with reviewer 2 that more orthogonal binding assays would initiate a deeper analysis of interesting candidates found in our data set. This we have done for HIP1 and could show that it is a real binder. However, we would like to point out that binding assays like biolayer interferometry (BLI) or surface plasmon resonance (SPR) (as we performed for HIP1) require large protein input for titration curves. Ubiquitin chain enzymatic synthesis is a multistep process of protein purification, chain synthesis and further purification which is time intensive and requires a large amount of protein input for a smaller protein yield, than in the purification of a single recombinant protein. For this reason, we were limited in the number of SPR measurements we could perform by amount of Ub chains we could reasonably synthesise. Therefore, it would not have been feasible to test the Ub binding of many different proteins to the 5 different types of Ub we used

(Ub, K48-linked Ub2, K48-linked Ub3, K63-linked Ub2, K63-linked Ub3 and K48/K63-linked branched Ub3), in order to validate their chain specificity.

Differences observed when either NEM or CAA was used to inactivate DUB activities in lysates feature prominently in the paper, but the source (and reliability) of those differences is unclear. Whereas the authors attribute the differences to modification of (poly)Ub interactors at the ubiquitin-binding interface, that was never established; rather, different extents of polyUb hydrolysis remain a plausible explanation. The authors contend - but do not show - that the immobilized polyUb bait exceeds the binding proteins in the lysates. In principle, endogenous DUBs in the lysates could cleave Ub from the biotinylated peptide tail that tethers it to the resin. Also, even limited hydrolysis of the branched K48/K63 polyUb could specifically enrich for either K63 or K48 diUb and bias the pulldown results. The authors note (p. 4) for Fig. 2 that among the many dozens of proteins in Cluster 1, only 4 appear in both the NEM and CAA datasets; to me, differential modifications of the ubiquitin-binding interfaces seems very unlikely as the explanation. In light of the emphasis given to the differences in the results with NEM vs CAA, they need further exploration. HIP1 was recovered in the branched K48/K63 polyUb pulldown treated with CAA but not NEM. As HIP1 binding was assayed *in vitro* with recombinant proteins (Fig. 5), the authors could test if treatment of recombinant HIP1 with either reagent affects the binding.

Reviewer 2 expressed concern that we attributed the differences between the NEM and CAA enrichment patterns to the 'differences to modification of (poly)Ub interactors at the Ub-binding interface', whilst they rightly point out that 'different extents of polyUb hydrolysis remain a plausible explanation'. We agree with the referee's opinion and would like to point out that we did clearly state both of these possibilities (lines 178-180): "These observations may be a result of increased chain stability by the more potent DUB inhibitor NEM or unspecific alkylation affecting Ub-binding sites, as is the case for IKBKG/NEMO²"

We were surprised, like reviewer 2, about the difference between CAA and NEM treated samples. We were also tempted to investigate that in greater detail. However, we are convinced that our presented results already provide a lot of information. Reviewer 2's suggestion to investigate the effect of CAA and NEM on HIP1 Ub-binding *in vitro* is experimentally feasible, however it does not fit the purpose of deciphering whether the modification of Ub-binding surfaces or Ub chain disassembly is responsible for different enrichment patterns. This is because in this particular case HIP1 is not detectable in any Ub chain pulldown under NEM treatment (as shown in the MS experiment and reproduced by Western Blot Figure 5A), despite NEM treatment leading to less chain disassembly than CAA treatment. Whereas, HIP1 binds Ub chains, with a preference for K48/K63-linked branched Ub3, under CAA treatment and *in vitro*. Thus, in this circumstance one can conclude that chain disassembly does not account for the reducing Ub-binding under NEM conditions. Furthermore, Pashkova et al show that a leucine and an aspartate residue are crucial for Ub-binding in the ANTH UBD³. In the ANTH domain of HIP1 this leucine (L250) is preceded by a cysteine (C249), making this cysteine a key candidate for potential alkylation by NEM or CAA, possibly abrogating Ub-binding. We added this detail to the discussion (lines 466-468).

The recent paper by Lange et al. (ref. 14) also identifies binders that prefer branched K48/K63 polyUb. Waltho et al. indicate a few cases where their results agree, but more often the results disagree; why is not discussed.

In agreement with reviewer 2, we find it extremely relevant for our readers to cite the preprint, and point out common hits. However, as it is a preprint, an in-depth comparison seems premature. We can, however, speculate on a few reasons for the differences - longer homotypic and K48/K63-linked branched chains used, different cell type, different pulldown conditions and different type of statistical comparison used. Additionally, whilst we did not find it appropriate to point out in the text, many of the proteins which they identified as 'branch-specific', but we did not, they could not validate by *in vitro* recombinant protein pulldown. Some did not bind Ub (ROCK2, ZFAND2B, PRKCZ) or equally bound to homotypic and branched Ub chains (DNAJB2, R1OK3) (Supplementary Figure 2C, Lange et al)⁴. Whereas, one of our two common hits, RFC1, was the only protein for which they were able to successfully validate branch-specificity by *in vitro* pulldown with a recombinant RFC1 (190-246) fragment (Figure 2F, Lange et al).

Finally, many aspects of the manuscript need clarification and other improvements. Important experimental details are lacking (e.g., how the recombinant HIP1 was made; what is the sequence of the "C-terminal" Cys-containing Ub protein). UBP is generally used to indicate a large subset of yeast DUBs; the use of "UBP" by Waltho et al. to signify ubiquitin binding protein is confusing. Most of the figures need more detailed legends. Ubiquitin has a C-terminal carboxylate, not "hydroxide" (p. 1).

We responded to reviewer 2's request for further clarification of experimental details. We added a section to the methods with the preparation of HIP1 (22-309)-biotin for SPR (lines 723-748) and corresponding plasmids

to the plasmid table. In order to clarify the placement of the cysteine in the proximal Ub we adjusted the wording in the methods section (lines 493-495) and in the results (lines 98-101).

We acknowledge that UBP is a yeast DUB family and thus have changed our acronym to ubiquitin-binding protein (UbBP).

Finally, we also changed the hydroxide error in the introduction (lines 42-44).

Referee #3:

Ubiquitylation is an important posttranslational modification that can determine the fate of modified proteins based on the linkage type between ubiquitin molecules in polymeric conjugates. The specificity of this ubiquitin code remains incompletely understood, as differences between chain types are sometimes quantitative in nature. Moreover, the consequences of ubiquitylation can also be determined by the number of ubiquitin molecules incorporated into a polymer, but how chain length is read out in cells, and when this is important, is not really known. Studies on further understanding how ubiquitin conjugates are read out and translated into biological reactions are therefore important.

In this manuscript, the authors address these questions by dipping ubiquitin dimers or trimers into cell extracts and determining bound proteins by mass spectrometry. This is not necessarily a novel approach, and previous investigations in this direction resulted in limited new knowledge about linkage-specific ubiquitin signaling. I am afraid the same is true for this study. For most of the study, the current approach does not discriminate between direct or indirect binders, a question that is important because a protein might be part of a complex that contains multiple ubiquitin recognition modules with distinct linkage specificity; such proteins might imply apparent preferences for a particular chain type (for example branched chains), while in reality their binding behavior reflects a combination of multiple independent binding events of the complex to different components of the conjugate. Moreover, lysates take proteins out of their natural context, i.e. a protein with apparent specificity for a particular linkage type or length might never encounter such chains if the respective enzymes and effectors do not localize to the same sites. Thus, at least some follow-up in cells is needed to show that anything documented here occurs without disrupting cell integrity and to show that any such interaction would be important. Third, whether a small preference seen at one concentration of a particular chain type is meaningful in cells depends on local concentrations of the modified protein and its effector; without knowing K_d values, concentrations of both partners in cells, and potential complex formation (i.e. local concentrations), the subtle preferences documented here might be an overestimate of the effects of linkage type or chain length. Finally, the vast majority of conjugates contain more than two or three moieties, but if principles uncovered for the minimal building blocks investigated here are important in the natural context of a more complex conjugate is not known. Given these significant issues, the current study appears to be rather preliminary.

Our approach is similar to that other scientists have chosen. Amongst them is the impactful resource paper by Zhang et al. (2017) which was highlighted with a preview article in *Molecular Cell* and has 138 citations to date. Using a similar Ub interactor pulldown method with chemically synthesized Ub chains of different linkage, Zhang provided the strongest and largest dataset of linkage-specific Ub interactors to date⁵. Our screen is novel because we use enzymatically synthesized Ub chains and compare K48- and K63-linked Ub chains of different length and branched Ub chains. We thus are convinced that our data set provide major new and important results which will be used by other laboratories.

Reviewer 3 correctly highlights the limitations of pulldowns. Like reviewer 1, reviewer 3 is also concerned about the ability to discern between direct and indirect binders. As outlined above, we addressed this in our discussion section (lines 475-478) and added a more detailed comparison of UBD-containing proteins in our data. For more details see response to reviewer 1 above. We understand that reviewer 3 would like to know the results of follow up experiments in intact cells to show that these Ub-interactor interactions exist and decipher the biological importance. However, this is out of the scope of a resource paper that provides data sets to other researchers. We would welcome further experiments from our readers.

Reviewer 3 claims that 'the vast majority of (Ub) conjugates contain more than two or three Ub moieties'. Whilst this may be the case *in vitro*, in yeast Ub chains exist mainly from dimer to heptamer length, with most K63-linked chains as dimers and most K48-linked chains as trimer and tetramers⁶. We added this extra detail to our reference to this study in the results section (lines 274-277). We would like to note that the exact complex architecture and length of Ub chains *in vivo* is a major knowledge gap in the Ub field. As a large screen for Ub interactors, our study does not attempt to answer that question, but instead works with a simplified, experimentally-viable system. We believe that this is the role of a screen.

Additional points:

1. I did not find information where exactly the Cys residue was introduced "near the C-terminus of ubiquitin". If this includes residue 74, which is often recognized by ubiquitin-binding proteins, it might inactivate an important site that contributes to chain length-dependent binding. This would complicate the interpretation of these results. Please provide this information in the main text of the paper. If Arg74 was used, please also repeat at least some experiments with an internal Cys mutant of ubiquitin that is distant from the I44 patch to ensure that adding biotin to this site does not interfere with recognition (alternatively, add some small protein and attach biotin to this protein).

In response to reviewer 2 and 3's request for further clarification of the placement of the cysteine in the proximal Ub, we added detail to the results and methods section (lines 493-495 and lines 98-101). As the cysteine has been placed in a linker attached after the C-terminus of the Ub, reviewer 3's concerns, whilst valid, do not apply to our method.

2. Fig. 3E/F needs inputs and bait-immobilized ubiquitin. This experiment provides little support for chain length- or linkage-specificity, as it is unclear how much ubiquitin was immobilized and how much protein was present in lysate.

In the legend for Figure 3 E and F, we clearly stated that the Western Blot is from the same pulldown experiment as in Figure 1B and therefore the input Ub bait silver stain in Figure 1B is also the input for Figure 3E and F. In order to make this clearer we now added the input Ub bait silver stain from Figure 1B to Figure 3 and combined E and F into one part, E.

3. It is generally disappointing how little follow-up was done from the mass spec. To really state that a protein has a preference for a certain linkage or chain length, quantitative binding experiments (for example by either fluorescence polarization, FRET, or SPR) would be required. Only K_d values determined by these methods would provide convincing information about binding specificity for a particular chain length or linkage. This is a critical issue, as the authors ascribed specificity based on relatively small differences in z scores that might not be relevant dependent on expression levels in cells, accessibility to a substrate in cells etc. This has been done for HIP1, where the data is nice, but would really have to be repeated much more throughout the paper. Also, some sort of validation in cells would be required. This would likely require much more work, such as identification of the responsible ubiquitin binding domains, mutations in this domain that reduce association with the preferred chain type and then analysis of any effects on ubiquitin binding in cells. I am not sure whether this could realistically be done within the context of a revision.

In response to reviewer 3's concern about the assignment of chain type specificity, we would like to draw attention to the careful language used in the manuscript in this regard. We only used 'specific' for previously determined chain-type specific UBPs and instead used 'preference' or 'enriched' to describe enrichment patterns in our data that have not been validated by orthogonal experiments or studies. In response to reviewer 3's comment on z scores we would like to point out that whilst the moderated F test and resulting heatmaps in our manuscript use z scores, we also performed two sample T tests to ascertain chain type preference.

As in our response to reviewer 2 and 3's concern for more orthogonal binding assays, we would like to point out that binding assays like biolayer interferometry (BLI) or surface plasmon resonance (SPR) (as we performed for HIP1) require large protein input for titration curves. Ubiquitin chain enzymatic synthesis is a multistep process of protein purification, chain synthesis and further purification which is time intensive and requires a large amount of protein input for a smaller protein yield, than in the purification of a single recombinant protein. For this reason, we were limited in the number of SPR measurements we could perform by amount of Ub chains we could reasonably synthesise. Therefore, it would not have been feasible to test the Ub binding of many different proteins to the 5 different types of Ub we used (Ub, K48-linked Ub2, K48-linked Ub3, K63-linked Ub2, K63-linked Ub3 and K48/K63-linked branched Ub3), in order to validate their chain specificity. Reviewer 3's suggestion of further experiments to identify novel UBPs and their biological relevance is outside the scope of this resource, but would be a possibility for our readers to exploit.

4. I am worried about their interpretation that branched ubiquitin chains contain information of each chain type. First, they see completely different correlations in each of their experimental systems (yeast vs. HeLa DUBin.1 vs HeLa DUBin.2). This suggests to me that correlations are driven by some idiosyncrasy of the experiment, not the biological system. Second, I am not sure that simply by looking at the branch point ubiquitin anyone should draw conclusions about signaling functions of branched chains, as most branched conjugates contain much longer and many more chain blocks (it is unlikely that branched chains contain a single branch point). Unfortunately, I believe that at the very least, much longer branched chains would need to be investigated, and substrates that are modified with them would also matter.

In response to reviewer 3's concern with the interpretation that branched Ub chains contain information of each constituent linkage type, we would like to refer to lines 348-352 which make this statement based on Figure 2C and D and Figure 4A-D, not Supplementary Figure 9A-C as the reviewer mentions. The clustered heatmaps in Figure 2C and D show that many proteins enriched on homotypic chains are also enriched on branched chains. In Figure 4A-D proteins labelled by linkage preference (two sample moderated T test K48

Ub3 versus K63 Ub3, adj.P < 0.05) also exhibited this linkage preference in a comparison with the branched chain.

Reviewer 3's claims that 'most branched chains contain much longer and many more chain blocks'. However, current literature has not been able to resolve the architecture of branched chains in cells. For example, Swatek et al and Crowe et al used MS-based techniques to detect the relative abundance of branched chains in cell lysate, however the architecture, including the length, of branched chains was not solved^{7,8}. Whilst we agree that branched chains are likely much more complex than the branchpoint alone, for our screen we chose the branchpoint as it is the minimal branch unit contained in any branched chain. Our aim was to address whether this feature alone confers UBP specificity. It has already been shown that the K11/K48-linked branchpoint binds more robustly to the proteasomal component RPN1 compared to a K48-linked Ub3 or mixed K11/K48-linked Ub3⁹. It is also relevant to note that in the preprint by Lange et al 'branch-specific' interactors were equally enriched on that branched Ub chains with either K48-branches on a K63-linked trimer or K63-branches on a K48-linked trimer⁴. Thus, these results support our argument for using the branchpoint. Reviewer 3 also rightly points out that the substrates of branched Ub chains may affect the signal of the branched chain, however this is outside of the scope of our screen.

5. Fig 5B: apparent preference for branched chains mostly reflects differences in how much ubiquitin had been immobilized to the beads.

We do not agree with reviewer 3's interpretation of the Western Blot in Figure 5B.

Minor point:

1. This manuscript would profit from proofreading by a native English speaker. There are several mistakes throughout the manuscript that are rather confusing ("we sort to expand..."). I apologize for pointing this out, but hope these mistakes could be corrected prior to publication.

The manuscript (written by a native British English speaker) has been proofread again and the mistake pointed out by reviewer 3 has been changed.

1. Nagaraj, N. *et al.* Deep proteome and transcriptome mapping of a human cancer cell line. *Mol Syst Biol* **7**, (2011).
2. Hooper, C., Jackson, S. S., Coughlin, E. E., Coon, J. J. & Miyamoto, S. Covalent modification of the NF- κ B essential modulator (NEMO) by a chemical compound can regulate its ubiquitin binding properties in Vitro. *Journal of Biological Chemistry* **289**, 33161–33174 (2014).
3. Pashkova, N. *et al.* ANTH domains within CALM, HIP1R, and Sla2 recognize ubiquitin internalization signals. *Elife* **10**, (2021).
4. Lange, S. M. *et al.* Comprehensive approach to study branched ubiquitin chains reveals roles for K48-K63 branches in VCP/p97-related processes. *bioRxiv* 2023.01.10.523363 (2023) doi:10.1101/2023.01.10.523363.
5. Zhang, X. *et al.* Proteome-wide identification of ubiquitin interactions using UbIA-MS. *Nature Protocols* **2018** 13:3 **13**, 530–550 (2018).
6. Tsuchiya, H. *et al.* Ub-ProT reveals global length and composition of protein ubiquitylation in cells. *Nature Communications* **2018** 9:1 **9**, 1–10 (2018).
7. Swatek, K. N. *et al.* Insights into ubiquitin chain architecture using Ub-clipping. *Nature* **572**, 533 (2019).
8. Crowe, S. O., Rana, A. S. J. B., Deol, K. K., Ge, Y. & Strieter, E. R. Ubiquitin Chain Enrichment Middle-Down Mass Spectrometry Enables Characterization of Branched Ubiquitin Chains in Cellulo. *Anal Chem* **89**, 4428–4434 (2017).
9. Boughton, A. J., Krueger, S. & Fushman, D. Branching via K11 and K48 Bestows Ubiquitin Chains with a Unique Interdomain Interface and Enhanced Affinity for Proteasomal Subunit Rpn1. *Structure* **28**, 43 (2020).

March 28, 2024

RE: Life Science Alliance Manuscript #LSA-2024-02740-T

Ms. Anita Waltho
Max Delbrück Center for Molecular Medicine
Robert-Rössle-Strasse 10
Berlin 13125
Germany

Dear Dr. Waltho,

Thank you for submitting your revised manuscript entitled "K48- and K63-linked ubiquitin chain interactome reveals branch- and length-specific interactors.". We would be happy to publish your paper in Life Science Alliance pending final revisions necessary to meet our formatting guidelines.

- please be sure that the authorship listing and order is correct
- please add a Running Title and a Summary Blurb/Alternate Abstract to our system
- please add ORCID ID for the secondary corresponding author -- they should have received instructions on how to do so
- please add the Twitter handle of your host institute/organization as well as your own or/and one of the authors in our system
- please note that the titles in the system and manuscript file must match
- the contributions selected for Volker Dötsch, Philipp Mertins, and Thomas Sommer do not qualify them for authorship. Please either update the contributions in our system and the Author Contributions section of the manuscript or let us know if the authors need to be removed.
- please add your main, supplementary figure, and table legends to the main manuscript text after the references section
- we encourage you to revise the figure legends for figures S2 and S12 such that the figure panels are introduced in alphabetical order
- please add a conflict of interest statement to your main manuscript text
- please add callouts for Figures S1H,I; S4D; S9D-F; S11A and S12A-G to your main manuscript text
- the proteomics data should be uploaded to an accessible database (PRIDE, for example) and the accession information provided in a Data Availability statement at the end of the Materials and Methods section

A. FINAL FILES:

B. MANUSCRIPT ORGANIZATION AND FORMATTING:

Sincerely,

May 8, 2024

RE: Life Science Alliance Manuscript #LSA-2024-02740-TR

Ms. Anita Waltho
Max Delbrück Center for Molecular Medicine
Robert-Rössle-Strasse 10
Berlin 13125
Germany

Dear Dr. Waltho,

Thank you for submitting your Resource entitled "K48- and K63-linked ubiquitin chain interactome reveals branch- and length-specific interactors.". It is a pleasure to let you know that your manuscript is now accepted for publication in Life Science Alliance. Congratulations on this interesting work.

DISTRIBUTION OF MATERIALS:

Again, congratulations on a very nice paper. I hope you found the review process to be constructive and are pleased with how the manuscript was handled editorially. We look forward to future exciting submissions from your lab.

Sincerely,
